# NusG inhibits RNA polymerase backtracking by stabilizing the minimal transcription bubble

**Matti Turtola, Georgiy A Belogurov\***

Department of Biochemistry, University of Turku, Turku, Finland

**Abstract** Universally conserved factors from NusG family bind at the upstream fork junction of transcription elongation complexes and modulate RNA synthesis in response to translation, processing, and folding of the nascent RNA. *Escherichia coli* NusG enhances transcription elongation in vitro by a poorly understood mechanism. Here we report that *E. coli* NusG slows Gre factor-stimulated cleavage of the nascent RNA, but does not measurably change the rates of single nucleotide addition and translocation by a non-paused RNA polymerase. We demonstrate that NusG slows RNA cleavage by inhibiting backtracking. This activity is abolished by mismatches in the upstream DNA and is independent of the gate and rudder loops, but is partially dependent on the lid loop. Our comprehensive mapping of the upstream fork junction by base analogue fluorescence and nucleic acids crosslinking suggests that NusG inhibits backtracking by stabilizing the minimal transcription bubble.

## Introduction

To control when and how fast genes are expressed, the activity of RNA polymerase (RNAP) is tightly regulated. Much of transcription regulation in all domains of life takes place at the initiation stage by modulating activities of promoters. The examples of on/off regulation at the transcript elongation stage, such as promoter-proximal pauses in metazoans (*Adelman and Lis, 2012*) and antitermination in prokaryotes (*Santangelo and Artsimovitch, 2011*), are also known. In other cases, transcription elongation control is mediated by coupling of transcription to downstream processes, such as RNA translation, processing, and folding (*Proshkin et al., 2010*; *Bubunenko et al., 2013*). The multisubunit RNAPs evolved to elongate relatively inefficiently in the absence of proper coupling, thereby enabling the downstream processes to control the elongation rate. The ubiquitous family of NusG proteins (SPT5/SPT4 in archaea and yeast, DSIF in mammals) are the central components which mediate coupling between transcription and the downstream processes (*Werner, 2012*).

The simplest member of the family, a bacterial NusG, consists of two domains connected by a flexible linker (*Figure 1*). The N-terminal domain (NTD or NGN) anchors NusG to the clamp helices of the RNAP β' subunit, whereas the C-terminal domain (CTD or KOW) interacts with the components of the downstream processes (reviewed in [*Belogurov and Artsimovitch, 2015*]). In *E. coli*, NusG CTD interacts with NusE as part of the ribosome on protein coding operons (*Burmann et al., 2010*) or as a part of a so-called antitermination complex (NusABEG) on ribosomal RNA operons (*Zellars and Squires, 1999*; *Shankar et al., 2007*; *Singh et al., 2016*); in these contexts, NusG inhibits the function of a transcription termination factor Rho. If neither ribosome nor antitermination complex is engaged, which often implies that transcription is futile, NusG CTD binds to Rho and facilitates termination of transcription (*Cardinale et al., 2008*; *Peters et al., 2012*). tRNA and other non-coding RNA genes escape the premature termination by Rho possibly due to their extensive secondary structures and small size relative to the transcribed region required for the termination of

**\*For correspondence:** gebelo@utu.fi

**Competing interests:** The authors declare that no competing interests exist.

**eLife digest** Cells decode genes in two steps. First, they synthesize a molecule similar to DNA, called RNA, which is a complementary copy of the gene. This process, known as transcription, creates an intermediate RNA molecule that is turned into protein in the second step. RNA polymerase is an enzyme that carries out transcription; it separates the two strands of the DNA helix so that the RNA can be synthesized from the DNA template. By opening up the DNA downstream of where active copying is taking place, and re-annealing it upstream, RNA polymerase maintains a structure called a "transcription bubble". RNA polymerases do not copy continuously but oscillate back and forth along the DNA. Sometimes larger backwards oscillations, known as backtracking, temporarily block the production of the RNA molecule and slow down the transcription process.

A protein called NusG helps to couple transcription to the other related processes that happen at the same time. One end of the protein, the N-terminal domain, anchors it to RNA polymerase and stimulates transcription elongation. The other end, the C-terminal domain, interacts with other proteins involved in the related processes and can positively or negatively control transcription elongation. Nevertheless it was poorly understood how NusG carries out these roles.

Turtola and Belogurov investigated how NusG from the bacterium *Escherichia coli* affects the individual steps of transcription elongation. A simple experimental system was used, consisting of short pieces of DNA and RNA, an RNA polymerase and NusG. A transcription bubble resembles an opening in a zipper with two sliders; and rather than affecting the synthesis of RNA, NusG affected the part that corresponds to the "slider" located at the rear edge of the bubble. NusG helped this slider-like element to bring the DNA strands at this edge of the bubble back together and modified it so that it behaved as a ratchet that inhibited RNA polymerase from backtracking. This did not affect the smaller backwards and forwards oscillations of RNA polymerase.

Turtola and Belogurov suggest that these newly discovered effects play a key role in regulating transcription; NusG's N-terminal domain makes the RNA polymerase more efficient, whilst the C-terminal domain makes it amenable to control by other proteins. Future studies will investigate whether these effects are seen in more complex experimental systems, which include proteins that interact with NusG.

transcription by Rho (*Mooney et al., 2009a*; *Peters et al., 2009*). NusG-mediated coupling of transcription with the pioneer round of translation is likely conserved in prokaryotes, whereas functioning of NusG CTD (and additional KOW domains present in eukaryotic SPT5 and DSIF) in RNA processing/maturation is likely conserved in all domains of life (*Belogurov and Artsimovitch, 2015*).

The regulation of gene expression by NusG-like proteins, which include several paralogs in some bacterial species, is complex. Even the housekeeping NusG may exhibit opposite effects on transcription in vivo depending on the protein partner(s) bound to its CTD domain. Furthermore, NusG from different bacteria display seemingly opposite effects on transcription by their cognate RNAPs in a purified in vitro system lacking the downstream components. The *E. coli* NusG has an intrinsic stimulatory effect on transcript elongation in vitro (*Bar-Nahum et al., 2005*; *Burova et al., 1995*), which persists when an isolated NusG NTD is used (*Mooney et al., 2009b*). It is hypothesized that this intrinsic stimulatory effect of NusG NTD may allow RNAP to transcribe more efficiently in vivo when coupled with the downstream processes and slower if the coupling is broken (*McGary and Nudler, 2013*; *Belogurov and Artsimovitch, 2015*; *Burmann and Rösch, 2011*). However, NusG from *Thermus thermophilus* slows down its cognate RNAP in vitro (*Sevostyanova and Artsimovitch, 2010*), whereas *Bacillus subtilis* NusG stimulates pausing by interacting with specific sequences in the non-template DNA (*Yakhnin et al., 2008, 2016*). We later suggest that these apparent incongruences result from the superimposition of several distinct consequences of the NusG NTD binding to the RNAP and considerable differences in the elongation properties of these RNAPs. But first we consider the mechanistic details of the elongation stimulation by *E. coli* NusG.

The RNAP nucleotide addition cycle consists of (*i*) the NTP substrate binding to a post-translocated transcription elongation complex (TEC); (*ii*) a chemical step of the nucleotide incorporation; and (*iii*) the post-catalytic relaxation of the resulting pre-translocated TEC, which involves the release

of pyrophosphate and forward translocation (reviewed in [**Belogurov and Artsimovitch, 2015**]). The processive repetition of this cycle is sometimes interrupted by off-pathway pause events. The latter can be classified into pauses involving backtracking of the RNAP on the DNA template by two or more registers, which occludes the active site with the nascent RNA (**Komissarova and Kashlev, 1997**; **Nudler et al., 1997**), and diverse non-backtracked pauses (**Artsimovitch and Landick, 2000**). The non-backtracked pauses involve more complex and less understood rearrangement of the active site and the RNAP structure that impede the nucleotide addition (**Hein et al., 2014**; **Zhang et al., 2010**; **Kireeva and Kashlev, 2009**). Notably, many non-backtracked pauses likely involve partial opening of the β'clamp (**Hein et al., 2014**; **Weixlbaumer et al., 2013**), a large mobile domain that contributes most of the β' subunit contacts with the nucleic acids.

The elongation-enhancing effect of NusG may arise from (*i*) accelerating the on-pathway elongation, (*ii*) diminishing some or all type of pauses, or both. Early studies suggested that NusG acts by reducing pausing (**Burova et al., 1995**). Recent reports further specified that NusG homologues enhance elongation by restricting the conformational flexibility of the RNAP β'clamp (**Sevostyanova et al., 2011**; **Hirtreiter et al., 2010**), which is consistent with biophysical measurements (**Schulz et al., 2016**) and structural considerations (**Martinez-Rucobo et al., 2011**; **Klein et al., 2011**). On the other hand, *E. coli* NusG mainly reduces the frequency and duration of backtracked pauses (**Herbert et al., 2010**) that are not explicitly known to involve the β' clamp

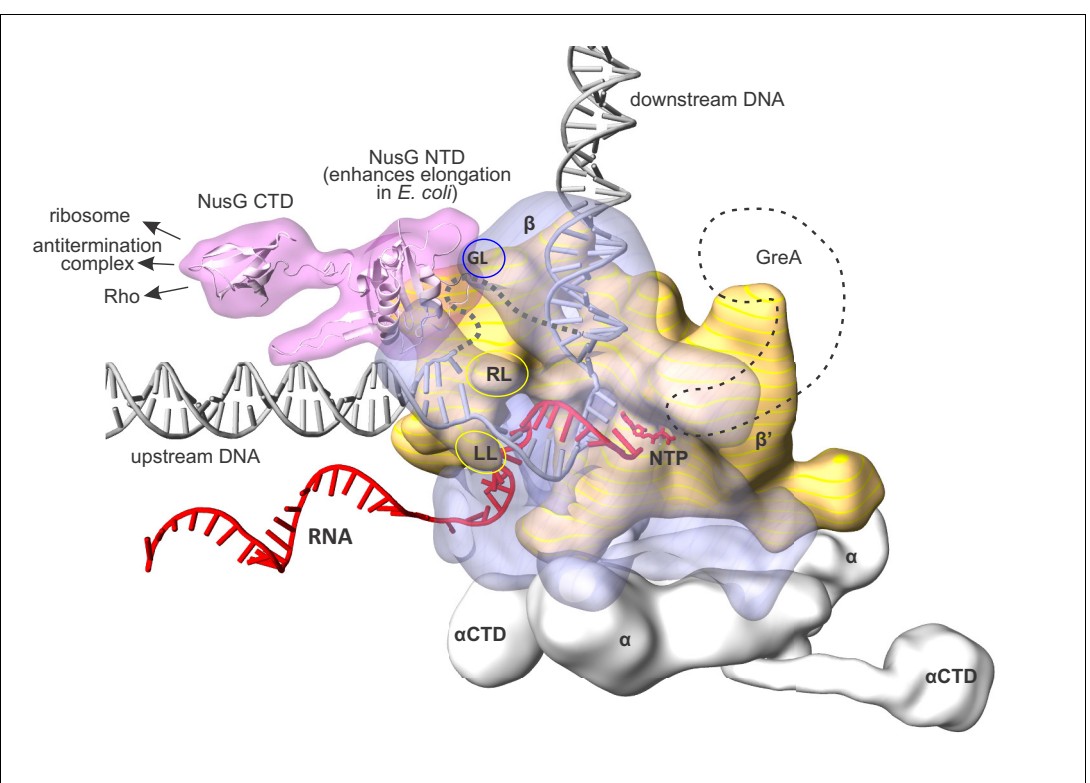

**Figure 1.** An overview of the bacterial transcription elongation complex (TEC) with bound NusG. RNAP core subunits are depicted by simplified differentially colored molecular surfaces. β is depicted transparent to reveal the path of nucleic acids through the enzyme. The positions of NusG CTD and αCTD, connected via flexible linkers, were chosen arbitrary. The locations of RNAP cleft loops individually deleted in this study, β Gate Loop (GL), β' Rudder Loop (RL) and β' Lid Loop (LL) are accentuated by ovals. The hypothetical path of the single stranded non-template DNA is depicted by a grey dashed line. The approximate location of GreA cleavage factor employed in backtracking experiments (see results) is depicted as a black dashed contour. The composite model was generated using the *Thermus thermophilus* TEC (**Vassylyev et al., 2007**), NusG NTD from the NusG-RNAP model in **Martinez-Rucobo et al., 2011** and the elements from other structures (see Materials and methods). The duplex DNA immediately upstream of the RNA:DNA hybrid was modeled *de novo* as described in the Results section.

opening (*Sekine et al., 2015*) and has only small stimulatory effect at non-backtracked pauses (*Artsimovitch and Landick, 2000*; *Belogurov et al., 2010*; *Kolb et al., 2014*).

Here, we report a detailed analysis of NusG effects on the individual steps of the nucleotide addition cycle, backtracking, and the conformation of the upstream DNA. Our results suggest that NusG slows backtracking without affecting the on-pathway elongation in non-paused TECs. We also demonstrate that NusG inhibits backtracking by restricting the melting of the upstream DNA, independently of the NusG-RNAP contacts that are important for stabilization of the β' clamp conformation. We further perform a comprehensive mapping of the upstream fork junction, determine the point of the upstream DNA reannealing, and provide a plausible mechanistic model of the anti-backtracking action of NusG.

## Results

### NusG does not affect the rates of nucleotide incorporation and translocation in a non-paused TEC

To determine the effect of NusG on the kinetics of nucleotide addition and translocation, we utilized a TEC design that was extensively validated in our previous studies (*Malinen et al., 2012*, *2015*). The TEC was assembled on a synthetic nucleic acid scaffold and contained the fully complementary transcription bubble flanked by 20-nucleotide DNA duplexes upstream and downstream (*Figure 2—figure supplement 1*). The annealing region of a 16-nucleotide RNA primer was initially nine nucleotides, permitting the TEC extended by one nucleotide to adopt the post- and pre-translocated states, but disfavoring backtracking. The RNA primer was 5' labeled with the infrared fluorophore ATTO680 to monitor the RNA extension by the denaturing PAGE. We performed parallel, time-resolved measurements of nucleotide incorporation by assembled TEC accompanied by forward translocation and, in a separate experiment, pyrophosphorolysis of extended TEC accompanied by backward translocation in the presence and absence of saturating concentration of NusG (2 μM, see later) (*Figure 2A–C*). RNA extension was monitored by a rapid chemical quench-flow method, whereas forward and backward translocation were monitored by measuring the fluorescence of the 6-methyl-isoxanthopterin base (6-MI) incorporated in the template DNA strand in a stopped-flow instrument (*Malinen et al., 2015*). NusG did not affect either of these on-pathway reactions. We then tested the effect of NusG on the TEC translocation bias in the equilibrium setup (*Figure 2D*). We have previously demonstrated that the predominantly post-translocated TEC can be converted into the pre-translocated state by tagetitoxin (TGT) (*Malinen et al., 2012*). TGT was equally potent in biasing the TEC towards the pre-translocated state in the presence and absence of NusG, suggesting that NusG does not affect the equilibrium between the post- and pre-translocated states. Overall, these experiments suggest that NusG does not measurably affect the on-pathway kinetics and thermodynamics of transcript elongation (*Table 1*) in the non-paused TECs examined in this study.

### NusG slows backtracking captured by GreA-mediated RNA cleavage

NusG has been suggested to inhibit stochastic and sequence-specific backtracking during transcription through a long template in vitro (*Herbert et al., 2010*). We developed a system where backtracking of the TEC is driven by the rapid cleavage of the nascent RNA by the RNAP active site. The reaction was initiated by adding 2–8 μM GreA protein that transforms the RNAP active site into a highly efficient RNAse. The TECs were assembled on a synthetic nucleic acid scaffold and contained the fully complementary transcription bubble (*Figure 2—figure supplement 1*). The annealing region of 18-nucleotide long RNA primer was 11 nucleotides, allowing the TEC to backtrack by one nucleotide. The RNA primer was 5' labeled with ATTO680 to monitor the accumulation of a 16-nucleotide RNA cleavage product in a rapid chemical quench flow experiment. The RNA primer also contained 2-aminopurine (2-AP) as the penultimate 3' nucleotide (*Figure 3A*), thereby permitting monitoring of 2-AP-p-C dinucleotide release by measuring an increase in 2-AP fluorescence in a stopped flow instrument. In a subset of experiments, the template DNA contained 6-MI nine registers upstream of the RNA 3' end (*Figure 3A*) to directly monitor RNAP backtracking by measuring the decrease in 6-MI fluorescence in a stopped flow instrument. Importantly, the changes in 6-MI and 2-AP fluorescence were driven by the RNA cleavage and not GreA binding because the addition

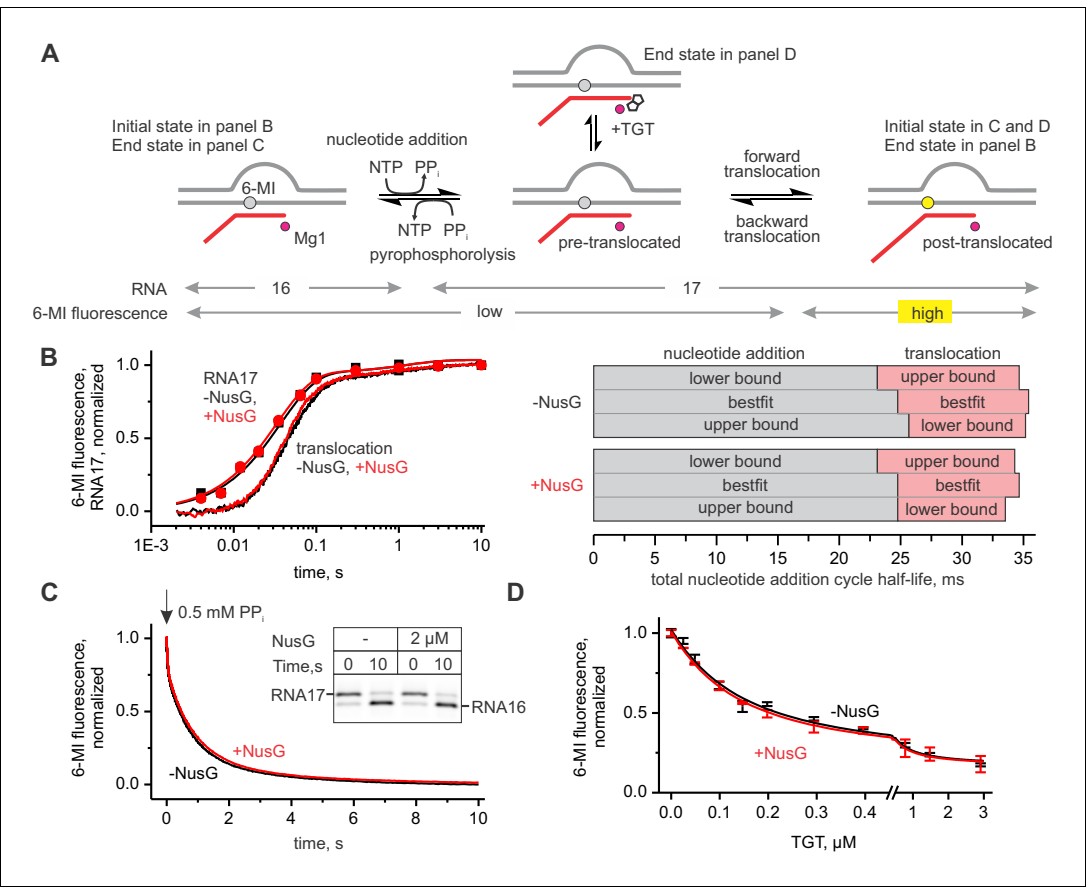

**Figure 2.** NusG does not affect on-pathway transcription elongation. (**A**) The schematics of the system used for monitoring the forward and the backward kinetics of the nucleotide addition cycle. In addition, the translocation bias of the TEC was evaluated under equilibrium conditions by measuring the TEC response to tagetitoxin (TGT). (**B**) The effect of NusG on the rate of nucleotide addition (discrete time-points and best-fit lines) and forward translocation (continuous time-traces). The lower and upper bounds of the reaction half-lives were calculated by combined analysis of data from several independent experiments (*Table 5*) by FitSpace routine of Kintek Explorer software (at a 10% increase in Chi2). (**C**) The effect of NusG on the pyrophosphorolysis-driven backward translocation. *Inset*: the gel control of the pyrophosphorolysis reaction. (**D**) The effect of NusG on the potency of TGT to bias the TEC towards the pre-translocated state. Error bars indicate the range of duplicate measurements. Numerical values of the reaction rate constants and the median reaction times are presented in *Table 1*.

The following figure supplement is available for figure 2:

**Figure supplement 1.** TEC systems employed in the experiments presented in the main figures.

of the cleavage-deficient D41N GreA variant did not change either fluorescence (*Figure 3—figure supplement 1*).

Parallel experiments demonstrated that within the limits of experimental uncertainty accumulation of the 16-nucleotide cleavage product, the release of 2AP-p-C dinucleotide, and RNAP backtracking occur with the same rate (*Figure 3B*). Importantly, 3' mismatched TECs that are more prone to backtracking were cleaved approximately thirty times faster than the matched TECs (*Figure 3C*), suggesting that for the matched TECs the rates of RNA cleavage and the dinucleotide release were limited by the rate of backtracking. We therefore concluded that all three types of measurements can be used interchangeably to monitor backtracking in this system.

Saturating concentrations of NusG (2 μM, see below) slowed RNA cleavage, dinucleotide release (increase in 2AP fluorescence), and backtracking (decrease in 6-MI fluorescence) approximately two-fold (*Figure 3B*). Similar inhibition was observed at 2 μM (*Figure 3—figure supplement 3*) and 8

**Table 1.** Numerical values of the reaction rate constants and the median reaction times.

| Figure | | | | | | | | |
|---|---|---|---|---|---|---|---|---|
| 2B | | | −NusG | | | +NusG | | |
| | | | lower bound | best fit | upper bound | lower bound | best fit | upper bound |
| | nucleotide addition, $s^{-1}$ | | 27 | 28 | 30 | 28 | 28 | 30 |
| | translocation, $s^{-1}$ | | 60 | 65 | 73 | 62 | 70 | 79 |
| | Slow TEC fraction | | | ~8% | | | ~7% | |
| | recovery rate, $s^{-1}$ | | 0.4 | 1.1 | 2.7 | 0.4 | 1.1 | 2.7 |
| | inactivation rate, $s^{-1}$ | | 0.03 | 0.09 | 0.3 | 0.02 | 0.08 | 0.2 |

The lower and upper bounds of rate constants were calculated by the combined analysis of data from several independent experiments (**Table 5**) by FitSpace routine of Kintek Explorer software (at a 10% increase in Chi2).

| | | | | | | |
|---|---|---|---|---|---|---|
| 2C | | median pyrophosphorolysis time, s | | | | |
| | | -NusG | | +NusG | | |
| | | 0.49 ± 0.08 | | 0.51 ± 0.08 | | |

Errors indicate the range of the bestfit estimates in duplicate experiments.

| | | | | | | |
|---|---|---|---|---|---|---|
| 2D | | $K_D$ TGT, µM | | | | |
| | | -NusG | | +NusG | | |
| | | 0.09–0.15 | | 0.09–0.14 | | |

The ranges represent 95% confidence interval for $K_D$ determined by the nonlinear regression analysis of data from two independent experiments.

| | | | | | | |
|---|---|---|---|---|---|---|
| 3B | Method | median reaction time, s | | | | |
| | | −NusG | | +NusG | | |
| | 6-MI | 13.2 ± 2.2 | | 28.0 ± 0.7 | | |
| | RNA18 | 11.7 ± 1.1 | | 30.4 ± 3.4 | | |
| | RNA16 | 11.6 ± 1.2 | | 30.1 ± 3.6 | | |
| | 2-AP | 12.4 ± 1.8 | | 30.3 ± 2.2 | | |

Errors indicate the range of the bestfit estimates in duplicate experiments.

| | | | | | | |
|---|---|---|---|---|---|---|
| 3C | RNA | DNA | RNAP | median reaction time, s | | |
| | | | | −NusG | | +NusG |
| | matched | matched | WT | 12.4 ± 1.8 | | 30.3 ± 2.2 |
| | matched | matched | ΔRL | 11.1 ± 1.9 | | 25.7 ± 4.4 |
| | matched | matched | ΔLL | 11.4 ± 1.3 | | 20.6 ± 3.1 |
| | matched | matched | ΔGL | 24.0 ± 2.3 | | 54.1 ± 4.7 |
| | matched | mm 1 | WT | 19.9 ± 2.2 | | 42.9 ± 3.3 |
| | matched | mm 1 and 2 | WT | 2.30 ± 0.04 | | 3.11 ± 0.20 |
| | 3' mm | matched | WT | 0.34 ± 0.17 | | 0.36 ± 0.10 |

3' mm stands for mismatch against 3' RNA nucleotide. Errors indicate the range of the bestfit estimates in duplicate experiments.

µM GreA (**Figure 3B**), suggesting that NusG did not act by weakening the binding of GreA. Furthermore, NusG did not slow the dinucleotide release from the TEC that was biased towards the backtracked state by the RNA 3′end mismatch (**Figure 3C**), suggesting that NusG did not directly inhibit RNA cleavage.

## NusG anti-backtracking activity requires the double-stranded upstream DNA

NusG may slow backtracking of the TEC by affecting the conformation of RNAP and/or transcription bubble. To dissect the mechanism of anti-backtracking activity of NusG, we individually deleted β

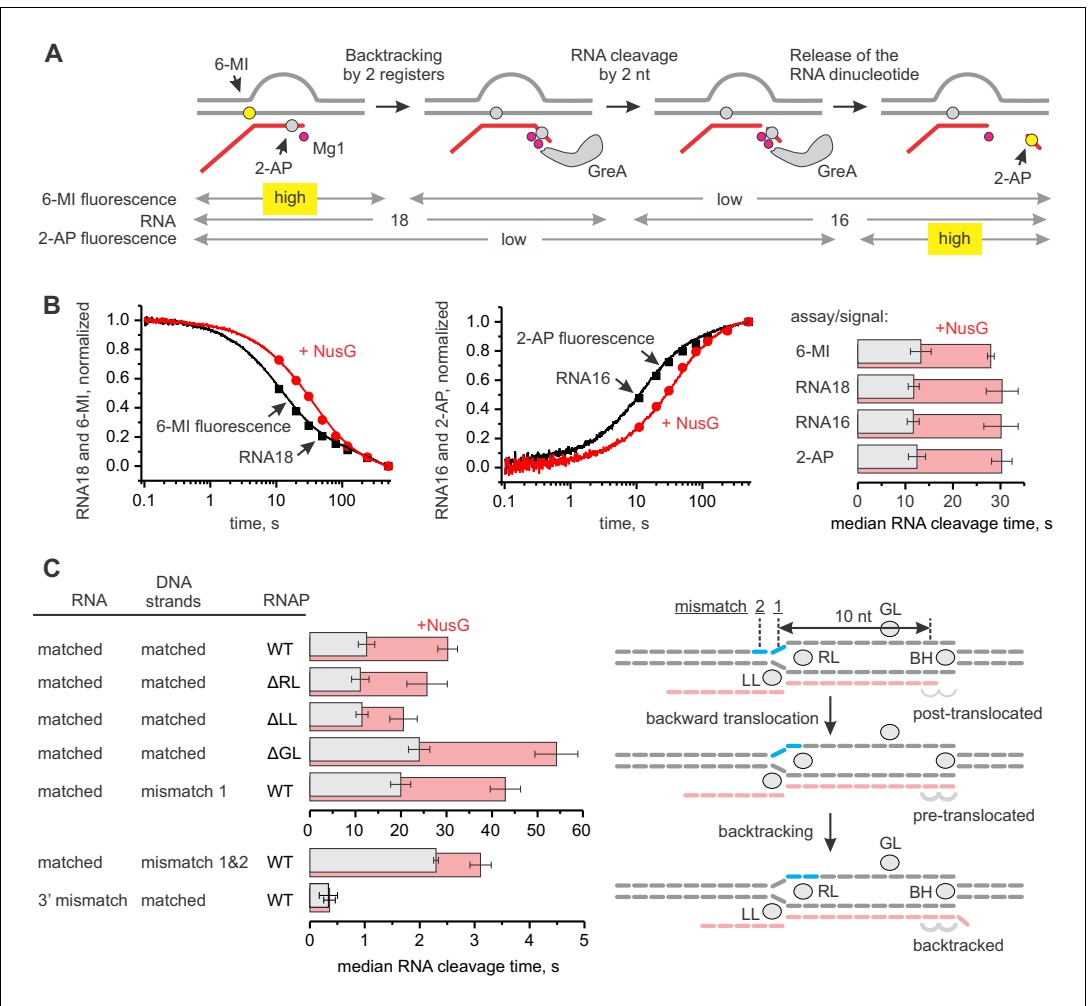

**Figure 3.** NusG inhibits GreA assisted RNA cleavage by slowing backtracking. (**A**) Three assays for monitoring the GreA-assisted RNA cleavage: TEC backtracking (6-MI fluorescence decrease), RNA cleavage (PAGE), and the dinucleotide release (2-AP fluorescence increase). (**B**) Left: the effect of NusG on the TEC backtracking (continuous time-traces) and RNA cleavage (discrete time-points) upon the addition of 8 μM GreA. Center: the effect of NusG on the release of the cleaved dinucleotide (continuous time-traces) and the RNA cleavage (discrete time-points). Right: the median reaction times. (**C**) The effect of NusG on the RNA cleavage by TECs with deletions of the RNAP domains, the mismatched upstream DNA and a 3' rCdA RNA:DNA mismatch. The median reaction times were determined by monitoring the increase in 2AP fluorescence at 8 μM GreA. The schematic on the right illustrates the location of DNA:DNA mismatches (cyan bars). Numerical values of the median reaction times are presented in *Table 1*. Error bars indicate the range of the bestfit estimates in duplicate experiments.

The following figure supplements are available for figure 3:

**Figure supplement 1.** Control experiment with the inactive GreA (D41N variant).
**Figure supplement 2.** Primary data for graphs in *Figure 3B*: The effect of NusG on GreA assisted RNA cleavage.
**Figure supplement 3.** The effect of NusG on RNA cleavage by wild-type TECs at 2 μM GreA.

and β' cleft loops near the NusG binding site (*Figure 1*). We evaluated the NusG effects on back-tracking of TECs assembled with RNAPs lacking β Gate Loop (ΔGL), β' Lid Loop (ΔLL) and β' Rudder Loop (ΔRL). We also perturbed the base pairs of the upstream DNA that may affect backtracking. While the register of the upstream DNA reannealing is not exactly known, the bacterial TEC

structures suggest that (*i*) DNA may reanneal as early as ten nucleotides upstream of the RNA 3' end in the post-translocated TEC and (*ii*) DNA must be unpaired up to at least 11 nucleotides upstream of the RNA 3' end in one-nucleotide backtracked TEC (*Figure 3C*). Accordingly, we evaluated the anti-backtracking activity of NusG on the TECs where DNA reannealing ten and ten-eleven nucleotides upstream from the RNA 3′end was inhibited by mismatches.

ΔRL and ΔLL did not affect backtracking rates in the absence of NusG, but the latter deletion reduced the TEC response to NusG twofold (*Figure 3C*). In contrast, ΔGL and the DNA:DNA mismatch ten nucleotides upstream of the RNA 3' slowed backtracking 1.5–2 fold but did not affect the TEC responses to NusG. Most notably, TEC with two DNA:DNA mismatches 10–11 nucleotides upstream of the RNA 3' end backtracked approximately fivefold faster than the fully-matched TEC and was insensitive to NusG (*Figure 3C*). At the same time, TEC with a 3' RNA:DNA mismatch cleaved RNA further sevenfold faster than the TEC with two DNA:DNA mismatches (*Figure 3C*), suggesting that backtracking limited the cleavage rate in the latter TEC. These results suggested that: (*i*) NusG slows backtracking by inhibiting DNA melting eleven nucleotides upstream from the RNA 3' end; (*ii*) RL, GL and DNA:DNA pairing ten nucleotides upstream of the RNA 3' end are dispensable for anti-backtracking activity of NusG; (*iii*) LL may be involved in the anti-backtracking action of NusG, but is not critically important therein.

## Mapping the effects of NusG on the TEC by DNA:DNA photocrosslinking with 8-methoxypsoralen

To directly test the effect of NusG on the reannealing of the upstream DNA, we developed a system to probe the DNA reannealing by photocrosslinking with 8-methoxypsoralen (8-MP). 8-MP specifically intercalates into the double-stranded 5′-TA-3′ sequence and introduces a T-T inter-strand crosslink upon illumination with UV light (*Figure 4—figure supplement 1*). We designed a fully complementary TEC containing a unique 5′-TA-3′ sequence motif positioned nine nucleotides upstream of the RNA 3' end (*Figure 4A*, *Figure 2—figure supplement 1*). The template DNA and the RNA primer were 5' labeled with ATTO680 to monitor DNA:DNA crosslinking and RNA extension by the denaturing PAGE. The system allowed us to probe DNA:DNA base pairing nine (TEC16), ten (TEC17) and eleven (TEC18) nucleotides upstream of the RNA 3' end by the stepwise extension of the RNA with subsets of NTPs (*Figure 4A*). The assembled TEC16 preparations produced only minute amounts of DNA:DNA crosslinked species as expected: the template DNA thymidine of TA site was anticipated to form the upstream-last base pair of the RNA:DNA hybrid. Upon formation of TEC17, the efficiency of DNA:DNA crosslinking remained low (<15%), despite the entire TA site being potentially available for pairing. In contrast, the crosslinking efficiency exceeded 40% in TEC18, comparable to that observed in a protein-free DNA:DNA duplex (~60%). TGT reduced crosslinking efficiency in TEC18 at least twofold (*Figure 4A*) consistent with its ability to stabilize TECs in the pre-translocated state (*Malinen et al., 2012*).

NusG effects on DNA:DNA crosslinking in the wild-type TEC18 were within the margins of the experimental errors (*Figure 4B*). However, NusG restored otherwise markedly reduced DNA crosslinking in ΔRL and ΔLL TEC18s. These observations suggest that NusG stabilizes the upstream DNA duplex 11 nucleotides upstream the RNA 3' end, but does not affect crosslinking in the wild-type TEC18 because the TA site is double-stranded even in the absence of NusG. Interestingly, NusG marginally but measurably enhanced crosslinking in the wild-type and ΔLL TEC17s, suggesting that NusG may affect the DNA conformation immediately upstream of the RNA:DNA hybrid (ten nucleotides upstream of the RNA 3' end).

## Mapping the effects of NusG on the TEC by RNA:DNA photocrosslinking with 6-thioguanine

In light of the effect of NusG on the upstream DNA reannealing, it was reasonable to test its effect on a related and spatially adjacent process of RNA:DNA separation. We used template DNA-RNA photo-crosslinking by a guanine analogue 6-thioguanine (6-TG), to probe the accessibility of RNA to DNA in the absence and presence of NusG. The TECs contained the fully complementary transcription bubble and the 5' ATTO680-labeled 16-nucleotide RNA primer with the nine nucleotide annealing region. The initial TEC16 contained 6-TG in the template DNA eight base pairs upstream of the RNA 3' end (*Figure 5A*, *Figure 2—figure supplement 1*). Upon exposure to UV light, the TEC16

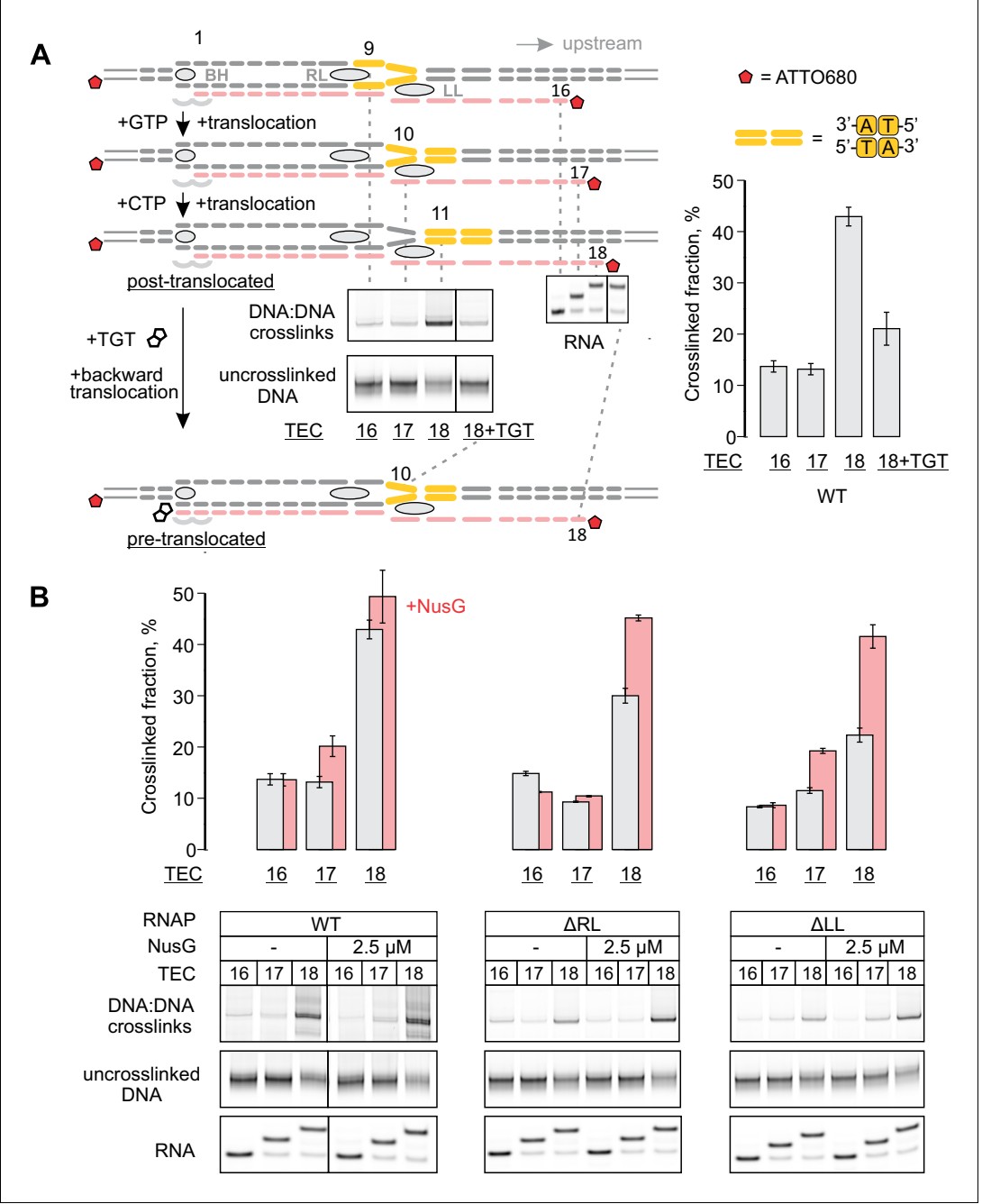

**Figure 4.** Probing the effects of NusG and deletions of the RNAP domains on the structure of the upstream fork junction by DNA:DNA photocrosslinking with 8-methoxypsoralen (8-MP). (**A**) TECs containing the unique 8-MP intercalation site were supplemented with 8-MP, walked by up to three nucleotides, supplemented with 5 µM TGT (where indicated) and illuminated with the UV light. (**B**) DNA:DNA crosslinking in TECs formed by the wild-type and altered RNAPs in the absence (grey bars) and presence (red bars) of NusG. Error bars indicate the range of duplicate measurements or SDs of several measurements (**Table 5**). The gel panels were spliced from the same gel and the pixel counts were linearly scaled to span the full 8 bit grayscale range within each panel. Joined panels have the same scaling.

The following figure supplement is available for figure 4:

**Figure supplement 1.** Control experiment: the specificity of DNA:DNA photocrosslinking with 8-MP.

produced crosslinked DNA-RNA species that migrated considerably slower than the un-crosslinked RNA primer in a denaturing gel (*Figure 5A*). Two major crosslinked species (a band and a smear) were observed that likely originated from 6-TG crosslinks to different RNA bases within the crosslinking range. Walking the RNAP along the DNA revealed that 6-TG efficiently crosslinks to the RNA primer eight (initial TEC16) and nine (TEC17) nucleotides upstream of the RNA 3' end. Crosslinking was largely abolished when the separation was increased to ten nucleotides in TEC18, yet restored when TEC18 was stabilized in the pre-translocated state by TGT (*Figure 5A*). Qualitatively similar results were obtained with the initial TEC16 containing 6-TG nine nucleotides upstream of the RNA 3' end (*Figure 5—figure supplement 1*); in this system, crosslinking was abolished upon an extension to form TEC17.

Deletion of the RL increased the overall intensity of crosslinks in TEC16 and TEC17, presumably by eliminating the protein domain that competed with RNA for crosslinking. More importantly, DNA efficiently crosslinked to RNA in ΔLL TEC18, but not in the ΔRL or the wild-type TEC18 (*Figure 5B*). These observations suggest that ΔLL makes the RNA accessible to DNA ten nucleotides upstream the RNA 3' end, but do not necessarily suggest that the RNA:DNA hybrid is longer in ΔLL TEC. An RNA with a mismatch against 6-TG efficiently crosslinked to DNA in the wild-type TEC17, demonstrating that the crosslinks reflect the physical accessibility of RNA to DNA and do not require the RNA:DNA base pairing (*Figure 5—figure supplement 1*). NusG did not measurably affect the crosslinking efficiency eight, nine or ten nucleotides upstream of the RNA 3' end in wild-type and altered TECs (*Figure 5B*), indicating that NusG does not alter the accessibility of RNA to DNA at the upstream edge of the transcription bubble.

## Mapping the effects of NusG on the TEC by a fluorescent beacon in the template DNA

We have previously reported that the base analogue fluorophore 6-MI positioned in template DNA strand within the 5'-TXG-3' beacon sequence (where X is 6-MI and G is a guanine functioning as a quencher) displays 2–5 fold brighter fluorescence when positioned nine and ten nucleotides upstream the RNA 3' end relative to fluorescence levels observed eight and eleven nucleotides upstream of the RNA 3' end (*Figure 6A*). This system was originally designed for the time-resolved studies of the RNAP translocation (*Malinen et al., 2012*). Here we revisit this setup to complement the photocrosslinking techniques in assessing the effects of NusG on base pairing immediately upstream of the RNA:DNA hybrid and on the conformation of the upstream DNA.

The TECs contained the fully complementary transcription bubble and 16-nucleotide RNA primer with nine nucleotides annealing region (*Figure 6A*, *Figure 2—figure supplement 1*). The initial TEC16 contained 6-MI base in template DNA positioned eight base pairs upstream of the RNA 3' end. The TEC was walked along the DNA by up to three positions by the addition of subsets of NTPs and the 6-MI fluorescence was monitored (*Figure 6A*). We attribute the low fluorescence of 6-MI eight nucleotides upstream the RNA 3' end (the assembled TEC16) to the quenching effect of the upstream guanine that forms the upstream-most base pair of the RNA:DNA hybrid. The elevated 6-MI fluorescence nine nucleotides upstream of the RNA 3' end is likely due to the unstacking of the upstream quenching guanine because it no longer belongs to the RNA:DNA hybrid. The elevated fluorescence persists ten nucleotides upstream of the RNA 3' end, arguing that 6-MI does not reestablish the stacking interaction with the upstream guanine. Finally, the 6-MI fluorescence is reduced 11 nucleotides upstream from the RNA 3' end, likely because 6-MI reestablishes the stacking interaction with the upstream guanine at this position. An experiment performed in the presence of TGT suggests that increased 6-MI fluorescence at the upstream edge of the RNA:DNA hybrid originates from a post-translocated TEC (*Figure 2D*).

The major effect of NusG on the wild-type TEC was the increase in the fluorescence of 6-MI at the upstream edge of the RNA:DNA hybrid (in TEC17). The effect was also observed in ΔLL and ΔRL TEC17s (*Figure 6C*, *Figure 6—figure supplement 1*). The rate of fluorescence increase in the wild-type TEC17 was dependent on NusG concentration with $K_d$ ~120 nM (*Figure 6C*). NusG binds about 30 Å from 6-MI at the upstream edge of the RNA:DNA hybrid and is therefore unlikely to affect the 6-MI fluorescence directly. Instead, we suggest that NusG increases 6-MI fluorescence by repositioning the quenching guanine immediately upstream of the RNA:DNA hybrid.

In the absence of LL, NusG significantly changed 6-MI fluorescence also in TEC18 and TEC19 (*Figure 6B*). Remarkably, ΔLL TECs deviated most from the wild-type TECs in the absence of NusG

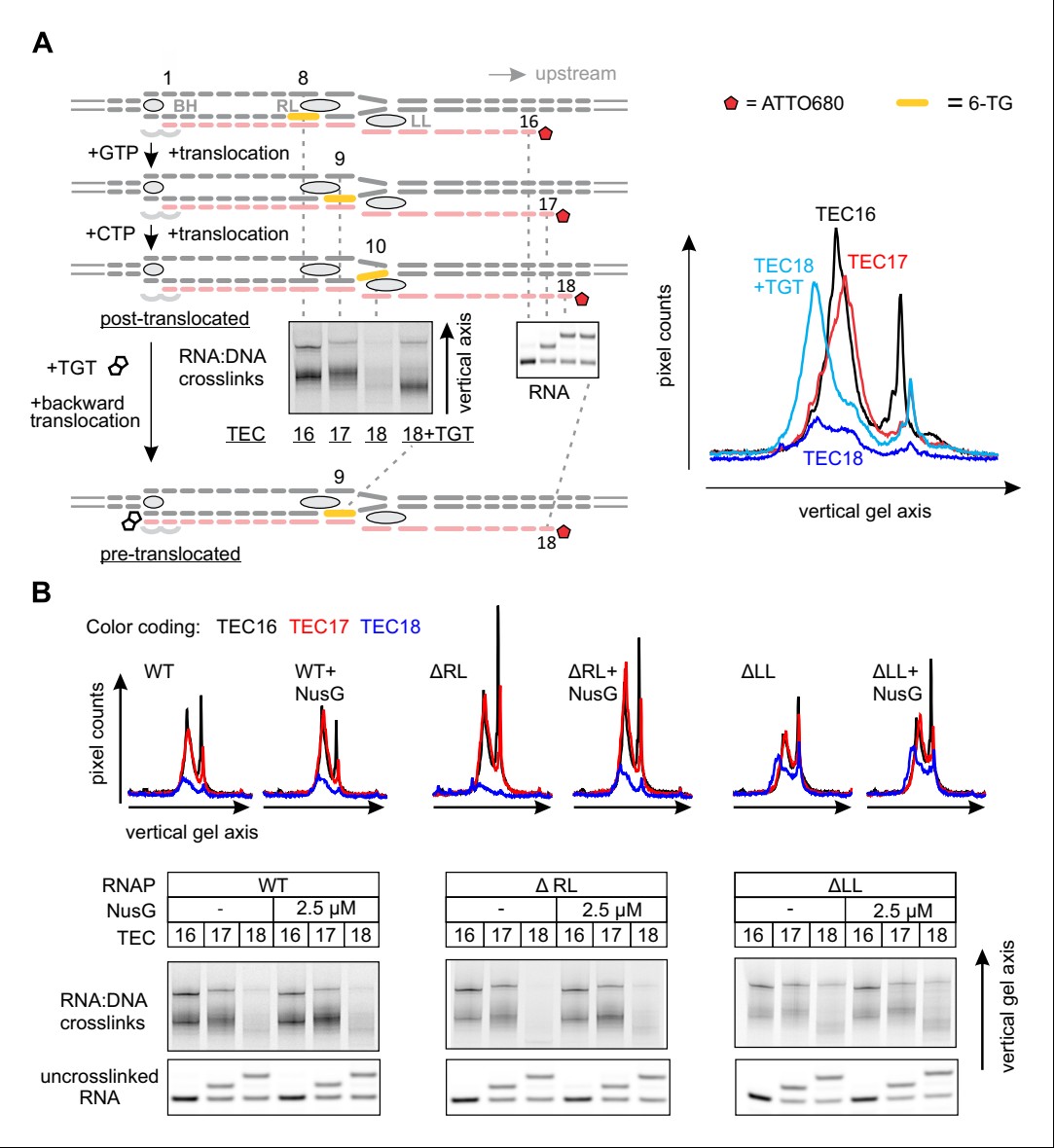

**Figure 5.** Probing the effects of NusG and deletions of the RNAP domains on the structure of the upstream fork junction by RNA:DNA photocrosslinking with 6-TG. (**A**) TECs containing 6-TG in the template DNA were walked by up to three nucleotides, supplemented with 5 µM TGT (where indicated) and illuminated with the UV light. (**B**) RNA:DNA crosslinking in TECs formed by the wild-type and altered RNAPs in the absence and presence of NusG. The gel panels were spliced from the same gel and the pixel counts were linearly scaled to span the full 8 bit grayscale range within each panel. The pixel intensity profiles for each gel lane are shown above the gels. The independent repeats are presented in *Figure 5—figure supplement 2*.

The following figure supplements are available for figure 5:

**Figure supplement 1.** RNA:DNA photocrosslinking with 6-TG reflects the physical accessibility of RNA to DNA.

**Figure supplement 2.** The effects of NusG (**A**), TGT (**B**) and ΔLL (**C**) on RNA:DNA photocrosslinking with 6-TG.

but displayed the identical fluorescence intensities in the presence of NusG (*Figure 6B*). Overall, the effects of NusG on 6-MI fluorescence largely paralleled its effects on the DNA crosslinking with 8-MP leading to similar conclusions: (*i*) NusG likely affects the DNA conformation immediately

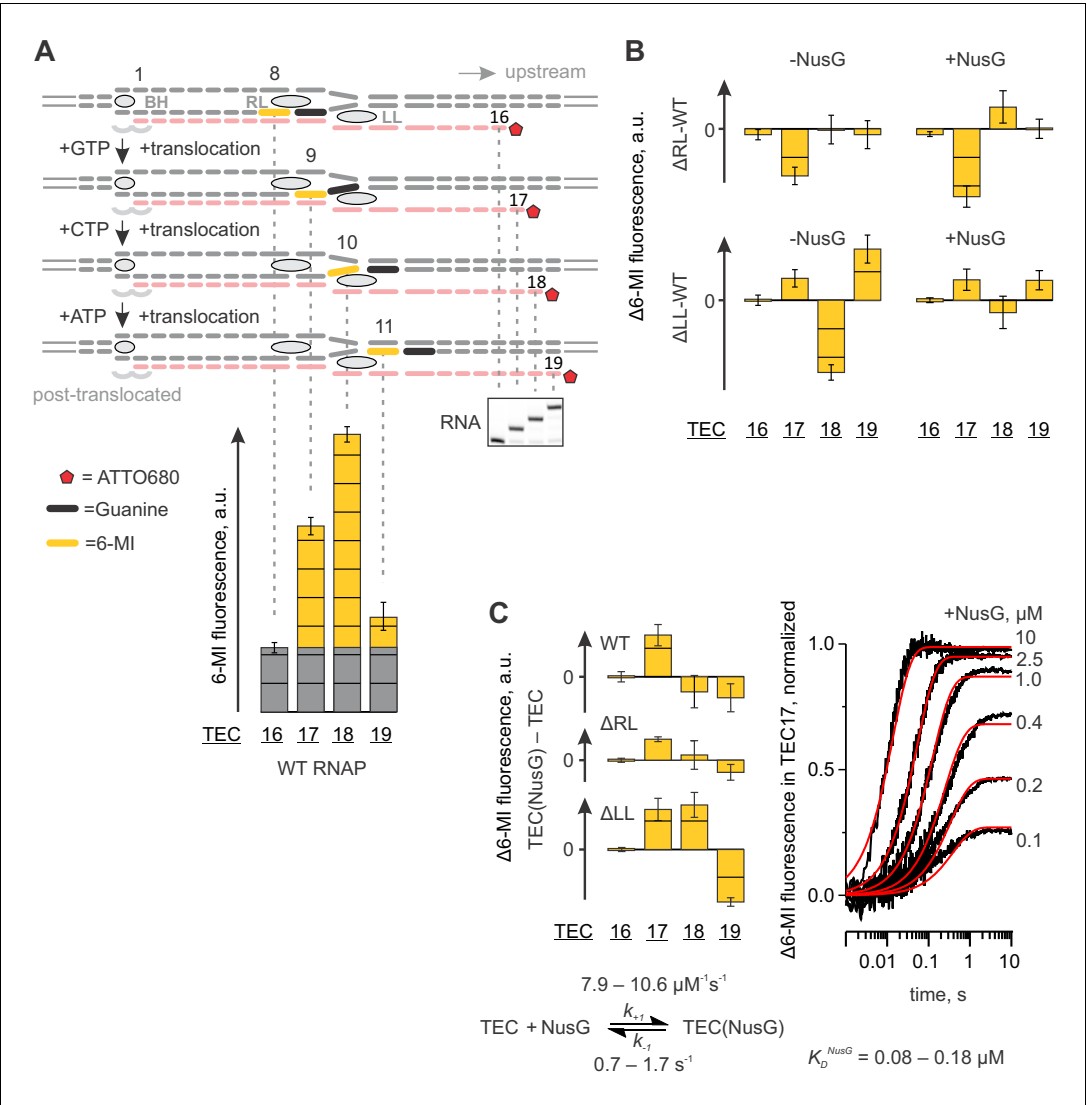

**Figure 6.** Probing the effects of NusG and deletions of the RNAP domains on the structure of the upstream fork junction by a fluorescent beacon in the template DNA. Error bars indicate the range of duplicate measurements or SDs of several measurements (*Table 5*). (A) Walking 6-MI (yellow dash) through the upstream fork junction modulates 6-MI stacking with the upstream guanine (black dash) that functions as a strong quencher. (B) The effects of ΔRL and ΔLL on the TEC fluorescence in the absence and the presence of NusG. (C) The effects of NusG on 6-MI fluorescence of the wild-type and altered TECs. Monitoring the fluorescence of TEC17 upon mixing with NusG in a stopped flow instrument (graph on the right, black curves) allows for the estimation of the binding and the dissociation rate constants. The analysis scheme is depicted below the graph. The best fit curves (red) were simulated using $k_{+1}$=9.2 μM$^{-1}$s$^{-1}$; $k_{-1}$=1.1 s$^{-1}$. The lower and upper bounds of rate constants were calculated by combined analysis of data from two independent experiments by FitSpace routine of Kintek Explorer software (at a 10% increase in Chi2).

The following figure supplement is available for figure 6:

**Figure supplement 1.** Primary data for *Figure 6*: the effects of NusG and deletions of the RNAP domains on 6-MI fluorescence.

upstream of the RNA:DNA hybrid and (*ii*) NusG reverses the alterations in the upstream DNA conformation introduced by deletion of the LL. However, in contrast to 8-MP crosslinking experiments,

NusG did not compensate for and, instead, increased the differences in the fluorescence intensities between the wild-type and ΔRL TECs (*Figure 6B*).

## An overview: NusG and the structure of the upstream fork junction

NusG likely inhibits backtracking by acting on the upstream DNA that, at the time of writing, is universally absent from the crystal structures of bacterial TECs. At the same time, the conformation of the upstream DNA in published TEC models (*Opalka et al., 2010*; *Martinez-Rucobo et al., 2011*; *Andrecka et al., 2009*) as well as recent X-ray (*Barnes et al., 2015*) and CryoEM (*Bernecky et al., 2016*) structures of RNA polymerase II are incompatible with the structure of bacterial TEC (in case of RNA polymerase II models), the upstream DNA mapping data presented here (see below), and/or NusG binding. Similarly, the conformation of the upstream DNA resolved in the crystal structures of the bacterial initiation complexes (*Zuo and Steitz, 2015*; *Bae et al., 2015*; *Liu et al., 2016*) is strongly influenced by the sigma factor and therefore is not suitable for modeling of the upstream fork junction in the TEC. To gain the mechanistic insights into the anti-backtracking action of NusG, we used our data to generate an accurate map of the upstream fork junction (*Figure 7B*) and further employed it to postulate a tentative structural model of a NusG-TEC complex with the upstream DNA (*Figures 1*, *8*).

Together, the 8-MP DNA:DNA photocrosslinking ( ± TGT), 6-TG DNA:RNA photocrosslinking ( ± TGT), 6-MI fluorescence pattern and crystal structures of bacterial TECs lacking the upstream DNA (*Vassylyev et al., 2007*) define the resting TECs as (*i*) post-translocated, (*ii*) containing nine base pairs RNA:DNA hybrid, and (*iii*) containing the upstream DNA duplex that starts eleven nucleotides upstream of the RNA 3′ end. The experiments with TECs containing mismatched non-template DNA strand additionally suggest that the upstream DNA base pairs ten nucleotides upstream of the RNA 3′ end (*Figure 7A*). Specifically, the 8-MP crosslinks the TA site eleven nucleotides upstream of the RNA 3′ end only when DNA is matched ten nucleotides upstream of the RNA 3′ end (*Figure 7A*, *Figure 7—figure supplement 2*). Similarly, TECs containing a DNA:DNA mismatch either directly against or one nucleotide upstream of 6-MI start to differ in fluorescence levels from the matched TECs when the mismatch is ten nucleotides from the RNA 3′ end (*Figure 7A*, *Figure 7—figure supplement 1*). Therefore, DNA likely base pairs ten nucleotides from the RNA 3′ end, but the first upstream base pair deviates from the geometry of the conventional B-form DNA duplex and/or is highly dynamic (*Figure 8B*).

To generate a NusG-TEC model, we positioned NusG NTD over the β′ clamp helices in bacterial TEC lacking the upstream DNA (*Vassylyev et al., 2007*) guided by the model of NusG-RNAP complex in (*Martinez-Rucobo et al., 2011*). We then modelled in the upstream duplex DNA following the overall direction suggested by the published models (*Opalka et al., 2010*; *Andrecka et al., 2009*; *Bernecky et al., 2016*) but avoiding clashes with the NusG NTD and maintaining a canonical B-duplex as far downstream as possible. Joining the upstream duplex with the RNA:DNA hybrid required altering the sugar-phosphate backbone of the downstream-most nucleotides of the upstream DNA but allowed maintaining the DNA base pairing. In the resulting model (*Figure 1*, *8*), the upstream DNA has an ample space to move away from the NusG NTD and the β′ clamp surfaces towards the cleft between the β1 and β flap domains by hinging around the downstream most base pair (position ten in *Figure 8*), reflecting the natural flexibility of the upstream DNA (*Coban et al., 2006*). However, we suggest that the conformation where the upstream DNA lines β′ clamp and NusG NTD (*Figure 1*, *8*) is the most relevant to the NusG effects on transcription elongation. In such a scenario, the two downstream- most base pairs of the upstream DNA (positions ten and eleven in *Figure 8*) occupy a narrow channel walled by NusG NTD and LL, thereby explaining the functional interactions between the NusG, the LL and the upstream DNA in RNA cleavage, crosslinking and fluorescence assays.

## Discussion

### NusG inhibits backtracking by acting on the upstream DNA

*E. coli* NusG was shown to enhance elongation in vitro over two decades ago, acting mainly by reducing pausing (*Burova et al., 1995*). More specifically, NusG was found to reduce backtracked but not hairpin-stimulated pauses (*Artsimovitch and Landick, 2000*; *Pasman and von Hippel,*

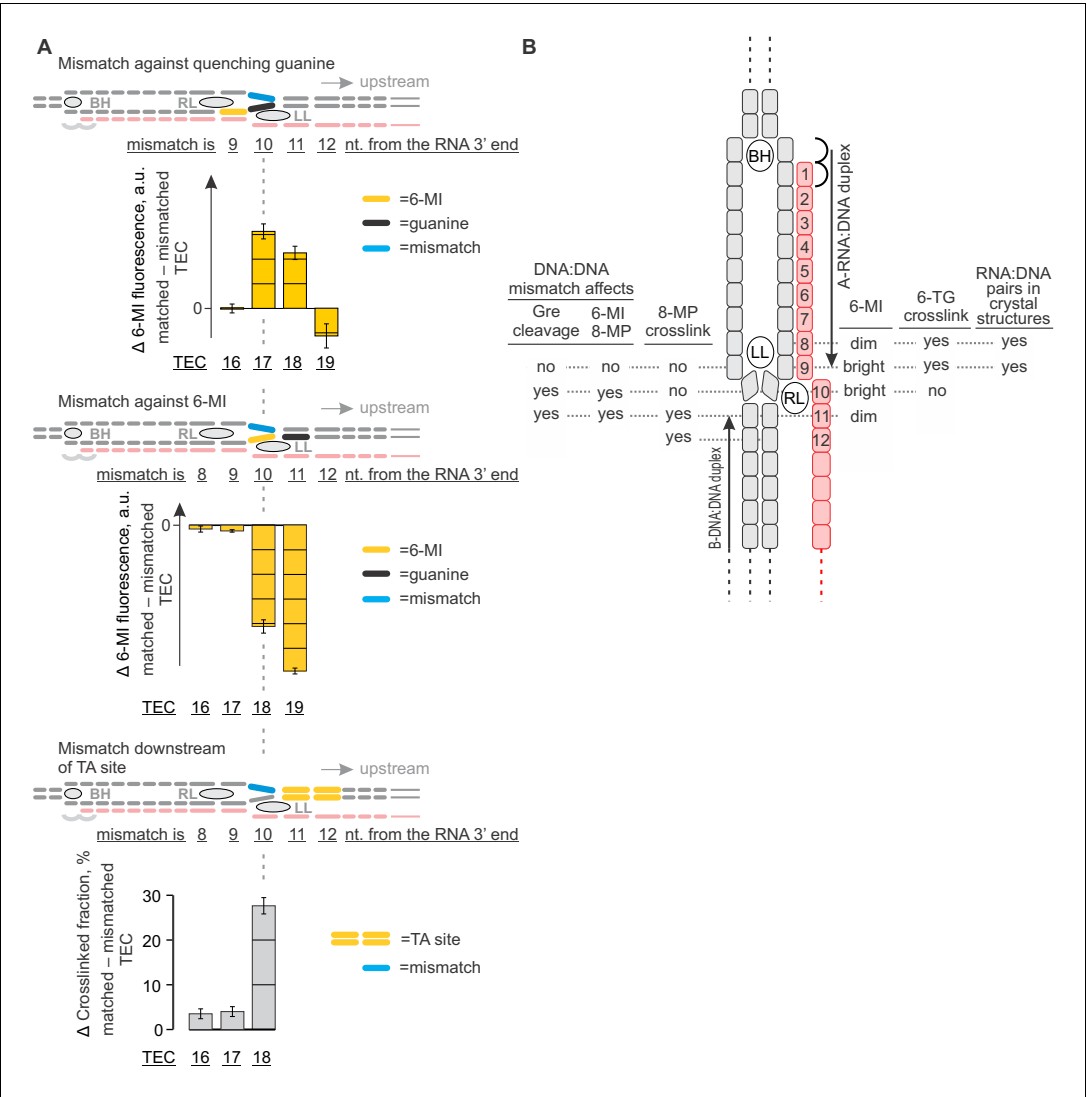

**Figure 7.** The effects of DNA mismatches suggest the minimal transcription bubble. (**A**) DNA:DNA mismatches against quenching guanine (top) and 6-MI (middle), or downstream of the TA site (bottom) alter the TEC properties when positioned ten nucleotides upstream of the RNA 3' end and further upstream. Error bars indicate the range of duplicate measurements or SDs of several measurements (**Table 5**). (**B**) Mapping the upstream edge of the transcription bubble based on data in **Figures 3C**, **4A**, **5A**, **6A** and **7A**.

The following figure supplements are available for figure 7:

**Figure supplement 1.** Primary data for **Figure 7A**: the effects of DNA mismatches on 6-MI fluorescence.

**Figure supplement 2.** Primary data for **Figure 7A**: the effect of a DNA mismatch downstream of the TA site on crosslinking with 8MP.

*2000*). It was later established that NusG-family proteins bind to, and bridge, the β' clamp with the β lobe across the RNAP cleft (*Belogurov et al., 2007*; *Mooney et al., 2009b*; *Klein et al., 2011*; *Martinez-Rucobo et al., 2011*). Biochemical studies further concluded that the archaeal NusG ortho-logue Spt5 (*Hirtreiter et al., 2010*) and the specialized NusG paralogue RfaH (*Sevostyanova et al., 2011*) enhance transcription elongation by stabilizing the β' clamp in a closed conformation. How-ever, *E. coli* RfaH accelerates RNAP at pause sites known to involve clamp opening, as well as at the backtracked pauses, whereas *E. coli* NusG has only marginal effect at the former sites (*Kolb et al.,*

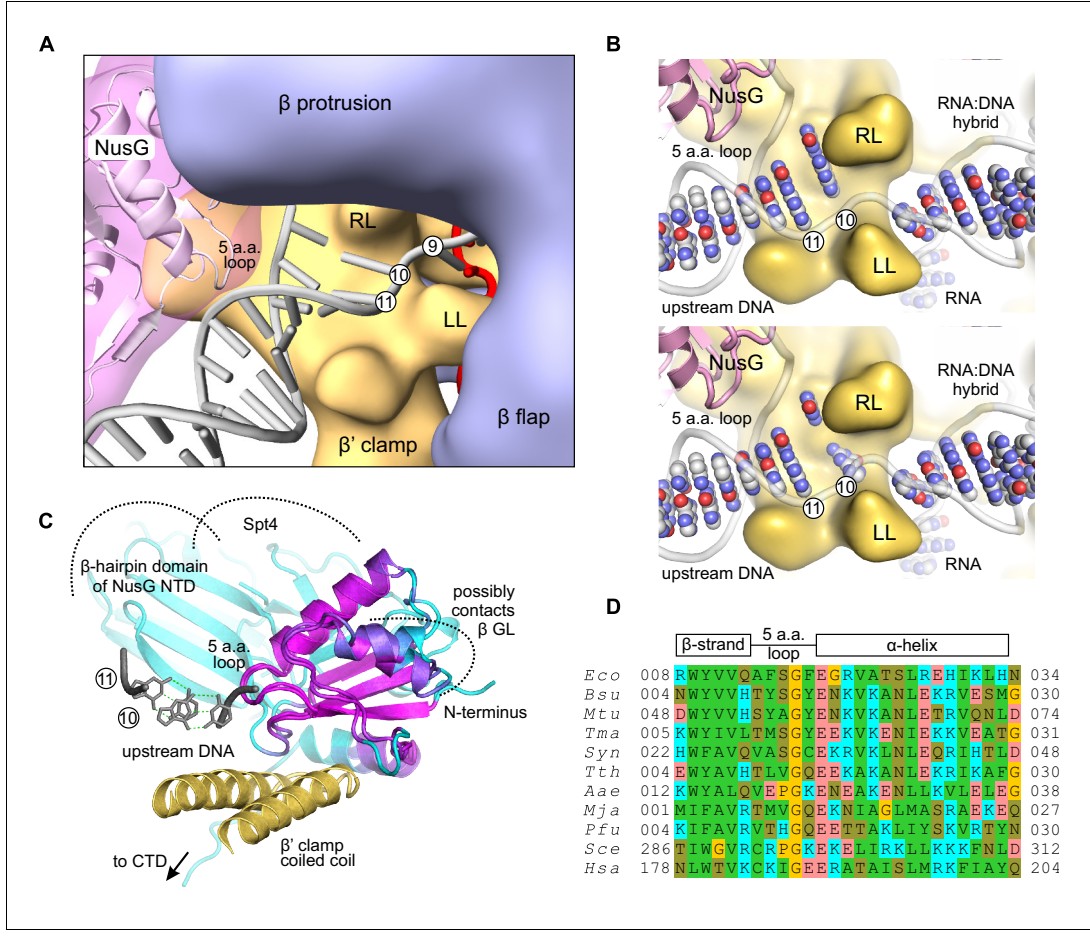

**Figure 8.** A model of the upstream fork junction. DNA bases are numbered from the RNA 3' end in the post-translocated TEC. (**A**) An overview: β' LL and the structurally conserved five amino acid loop of NusG NTD form a channel accommodating the exiting upstream DNA. (**B**) The template DNA nucleotide at position ten can be modeled to pair with the non-template DNA in a partially unstacked conformation (top) or interact with the cleft between the β' RL and β' LL (bottom). DNA and RNA bases are shown as spheres, sugar-phosphate backbones as cartoons. The β subunit is omitted for clarity. (**C**) The superimposition of *Aquifex aeolicus* NusG NTD (PDB ID 1M1G) and *Pyrococcus furiosus* Spt5 NTD (PDB ID 3P8B). The cartoon color changes from magenta in well superimposed regions to cyan in poorly superimposed or unaligned regions. The β' clamp helices (yellow, from PDB ID 3QQC), the upstream DNA (grey, from the model in **A**) and Spt4 (cyan cartoon, from PDB ID 3P8B) are included to present the superimposition in the context of the TEC. (**D**) Multiple sequence alignment of the structurally conserved five amino acid loop of NusG family proteins and the flanking secondary structure elements. Species names are abbreviated as follows: *Eco, E. coli, Bsu, Bacillus subtilis, Mtu, Mycobacterium tuberculosis, Tma, Thermotoga maritima, Syn, Synechocystis sp. PCC 6803, Tth, T. thermophilus, Aae, Aquifex aeolicus, Mja Methanocaldococcus jannaschii, Pfu, Pyrococcus furiosus, Sce, Saccharomyces cerevisiae, Hsa, Homo sapiens.* Amino acid residues are shaded as follows: hydrophobic –green, polar –olive, Pro and Gly –yellow, Asp and Glu – red, Arg, Lys and His –cyan.

The following source data is available for figure 8:

**Source data 1.** NusG-TEC model.

*2014*; *Anthony et al., 2000*; *Belogurov et al., 2010*). Moreover, crystal structures of the back-tracked TECs revealed the closed clamp (*Sekine et al., 2015*; *Wang et al., 2009*), whereas the specificity of *E. coli* NusG for backtracked pauses was reaffirmed in single molecule experiments (*Herbert et al., 2010*). Together, these observations suggest that *E. coli* NusG enhances transcription elongation by means other than restricting the β' clamp movement.

Here, we show that NusG slows backtracking but does not affect the on-pathway elongation in the non-paused TEC used in our study. In contrast, Herbert et al (*Herbert et al., 2010*) concluded that NusG has a modest stimulatory effect (10–20%) on the pause-free elongation rate, in addition to inhibiting backtracking. One possibility is that a subset of TECs backtracked by one nucleotide display the elongation rates within the pause-free range compiled by Herbert *et al*. Alternatively, NusG may have a marginal effect on the elongation rate in a subset of the non-paused on-pathway TECs with yet to be identified sequence determinants. In any case, the effect of NusG on the pause-free elongation rate estimated by Herbert et al is small comparing with the specific effect of NusG on the backtracking rate (~2.5 fold) that we report here.

We further demonstrate that two DNA mismatches immediately upstream of the RNA:DNA hybrid increase the backtracking rate and render the TEC insensitive to NusG. In contrast, NusG reduced backtracking normally in a TEC missing the GL, a contact point with the β subunit (*Figure 1*) that is required for anti-pausing by *E. coli* RfaH (*Sevostyanova et al., 2011*). β' RL was similarly dispensable, whereas the β' LL was slightly stimulatory for the anti-backtracking activity of NusG. Overall, our data suggest that the intrinsic action of NusG on the *E. coli* TEC is restricted to inhibiting backtracking and is exclusively mediated through the upstream fork junction. Remarkably, several other transcription factors, including Mfd (*Deaconescu et al., 2006*) and UvrD (*Epshtein et al., 2014*) that link transcription to DNA repair operate through the upstream fork junction.

Interestingly, ΔGL TEC backtracked two-fold slower than the wild-type TEC both in the presence and in the absence of NusG (*Figure 3C*). GL is located more than 20Å from the duplex DNAs and the RNA:DNA hybrid but may directly contact the single-stranded non-template DNA. Accordingly, we suggest that GL promotes backtracking by altering the conformation of the non-template DNA in a manner that increases the propensity of the TEC to backtrack, e.g. by facilitating the downstream DNA re-annealing. Indeed, GL restricts the downstream portion of the single-stranded non-template DNA within the main channel in the initiation complex (*Zhang et al., 2012*) and may therefore have a similar functionality in the TEC.

## Mapping the structure of the upstream fork junction

The upstream DNA decisively emerged as the major determinant of NusG anti-backtracking effect. However, the upstream DNA is absent from the crystal structures of bacterial TECs, the published models are incompatible with NusG binding, and even the register of the upstream DNA reannealing is uncertain. To gain mechanistic insights into the anti-backtracking action of NusG, we performed a comprehensive mapping of the upstream fork junction using fluorescent base analogues and site-specific crosslinking. Our data are fully consistent with the nine base pairs RNA:DNA hybrid and the upstream DNA duplex that starts eleven base pairs upstream of the RNA 3' end in the post-translocated TEC (*Figure 7B*). The effects of DNA mismatches additionally suggest that DNA is paired immediately upstream of the RNA:DNA hybrid. In combination, the data obtained with the matched and mismatched TECs suggest that the first pair of the upstream DNA is unstacked from both the RNA:DNA hybrid and the rest of the upstream DNA.

We then combined our mapping data with the model of the NusG-RNAP complex (*Martinez-Rucobo et al., 2011*) to sketch a NusG-TEC model with the upstream DNA. We found that in a subset of spatially feasible conformations of the upstream DNA (see results), the two downstream-most base pairs are accommodated in a narrow 'exit' channel walled by the NusG and the LL (*Figure 1, 8*). Such an arrangement plausibly explains the cooperation between NusG and the LL in stabilizing the upstream DNA pairing and inhibiting backtracking. Notably, NusG loop facing the upstream DNA is strictly conserved in size in bacteria and archaea (five amino acids, *Figure 8*), but the evidence for the conserved residue-specific contacts between the NusG and the upstream DNA is lacking. We propose that NusG provides a complementary molecular surface to the paired upstream DNA and possibly also affects the overall direction of the upstream DNA duplex. Finally, NusG likely stabilizes DNA pairing only in a subset of the upstream DNA conformations, yet influences the overall backtracking rate by targeting those conformations that are most favorable for backtracking, *i.e.*, the non-template DNA is optimally positioned for the strand exchange with the RNA.

## NusG inhibits backtracking by stabilizing the minimal bubble in the pre-translocated TEC

NusG inhibits backtracking by stabilizing the DNA base pair eleven nucleotides upstream of the RNA 3' end (*Figure 3C*). This pair corresponds to the second DNA pair upstream of the RNA:DNA hybrid in the post-translocated TEC (pair eleven in *Figure 8*). However, NusG does not affect the equilibrium between the pre- and post-translocated states (*Figure 2C–D*) and therefore likely acts on the pre-translocated TEC where the base pair eleven nucleotides upstream of the RNA 3' end lies immediately upstream of the RNA:DNA hybrid (pair ten in *Figure 8*, see also schematics in *Figure 3C*). Accordingly, we propose that NusG facilitates DNA pairing immediately upstream of the RNA:DNA hybrid in the pre-translocated TEC, thereby reducing backtracking.

It remains uncertain how NusG inhibits backtracking without affecting the equilibrium between the post- and pre- translocated states (*Figure 2C–D*). One possible explanation is that backtracking and backward translocation are limited by different processes in our system. It has been hypothesized that, rather than moving in sync along the different nucleic acid chains of the TEC, RNAP moves forward by sequentially translocating the RNA:DNA hybrid and the downstream DNA (*Brueckner and Cramer, 2008*). The synchronous sliding of RNAP along the nucleic acids is only superficially plausible in the 2D schematics (*Figures 2–7*) but is much less likely in the actual 3D structure of the TEC (*Figure 1*) where the downstream DNA and the RNA:DNA hybrid are separated by a 90° bend. Analogously, the backward translocation may involve the sequential translocation of the downstream DNA and the RNA:DNA hybrid. The former process is NusG independent and may limit the rate of the backward translocation, whereas the latter process is modulated by NusG and may be thus rate limiting for backtracking. We further argue that the difficulty of reconciling the large effect of NusG on backtracking with its small (*Herbert et al., 2010*) or undetectable (this work) effect on the forward and backward translocation in the context of a single-step translocation model lends support to a two-step translocation mechanism.

## Three distinct effects of NusG NTD on the TEC

The available data suggest three independent and structurally plausible effects of NusG NTD on the TEC. First, NusG binds near the upstream fork junction and stabilizes the upstream DNA duplex, thereby inhibiting spontaneous backtracking at most template positions ([*Herbert et al., 2010*] and this work). Second, the NTD restricts the conformational flexibility of the β' clamp, with different outcomes for the transcription elongation. Archaeal Spt5 (*Hirtreiter et al., 2010*; *Schulz et al., 2016*) and *E. coli* RfaH (*Sevostyanova et al., 2011*) exert at least part of their elongation enhancing effects by stabilizing the clamp. In contrast, clamp stabilization by *E. coli* NusG only marginally contributes to anti-pausing (*Kolb et al., 2014*; *Belogurov et al., 2010*). Third, NusG binds to specific sequences in the single-stranded non-template DNA, thereby introducing infrequent yet physiologically relevant pauses (*Yakhnin et al., 2016*, *2008*) and facilitating intrinsic termination in some species (*Czyz et al., 2014*). Similar effects are well documented for other dissociable factors and TEC components positioned near the single stranded non-template DNA (*Perdue and Roberts, 2011*; *Artsimovitch and Landick, 2002*; *Vvedenskaya et al., 2014*; *Arimbasseri and Maraia, 2015*). Interestingly, *T. thermophilus* NusG slows down the cognate RNAPs at non-paused sites by an unknown mechanism (*Sevostyanova and Artsimovitch, 2010*). It remains to be determined if this unusual effect is mediated through the clamp or contacts with the non-template and upstream DNA.

## Functional significance of the intrinsic stimulation of transcription elongation by NusG family proteins

We argue that anti-backtracking represents the only conserved functionality of NusG family proteins. Backtracking is a universally conserved and functionally important feature of the multisubunit RNAPs that has been documented in vitro and in vivo in both bacteria and eukaryotes (reviewed in [*Nudler, 2012*]). The stimulation of RNA chain elongation by NusG has been documented in vitro for bacterial (*Burova et al., 1995*), archaeal (*Hirtreiter et al., 2010*) and eukaryotic (*Wada et al., 1998*) transcription systems. The NTD is sufficient for these effects on elongation, but the structural elements that superimpose well between the bacterial NusG NTD and the archaeal Spt5 NTD (*Figure 8C*) are limited to (*i*) the beta sheet that comprises the RNAP-binding site, (*ii*) the conserved five amino acid loop that we implicate in the anti-backtracking action of *E. coli* NusG, and (*iii*) the

N-terminus of the α-helix that follows this loop and possibly interacts with the single-stranded non-template DNA (*Crickard et al., 2016*). The lack of the strong conservation of the surface residues (*Figure 8D*) suggests that the anti-backtracking activity may be determined by the overall fold of the NusG NTD and is only weakly dependent on the nature of the individual amino acid side-chains, consistent with the mutational analysis of *E. coli* NusG (*Mooney et al., 2009b*).

While the above considerations suggest that the stimulation of transcription elongation by NusG family proteins may be important for long-term survival and fitness, in the experimental systems studied to date, functional contacts established by the CTD, such as Rho and S10 in *Bacteria*, appear to be more critical. In *E. coli*, the essential function of NusG is to facilitate termination of transcription by Rho, thereby maintaining the operon borders (*Cardinale et al., 2008*), suppressing pervasive antisense transcription (*Peters et al., 2012*), and inhibiting R-loop formation (*Krishna Leela et al., 2013*). Stimulating Rho-dependent termination is also likely the major, albeit a non-essential, function of *B. subtilis* NusG (*Ingham and Furneaux, 2000*). NusG and its paralog RfaH have also been proposed to mediate transcription-translation coupling via direct contacts with S10 (*Burmann et al., 2010*, *Burmann and Rösch, 2011*). However, the CTD contacts are not universally conserved: Rho is absent in eukaryotes and even some *Bacteria* (*D'Heygère et al., 2013*), whereas S10 and RNAP are separated by a nuclear membrane in eukaryotes, where CTD interacts with proteins involved in splicing, polyadenylation, and other RNA processing pathways.

In eukaryotes, the intrinsic stimulatory activity of Spt5 NTD on transcription elongation is non-essential, but abolishing it leads to the temperature sensitive phenotypes (*Crickard et al., 2016*). In rare circumstances, the stimulatory effect of Spt4/5 may possibly be deleterious: Stp4/5 has been suggested to facilitate transcription through toxic repeat sequences in eukaryotes, thereby contributing to the progress of neurodegenerative disorders (*Kramer et al., 2016*). Overall, we suggest that the universal conservation of the intrinsic stimulatory activity of NusG family proteins on transcription elongation underscores its importance, but the quantitative assessment of the in vivo role of this functionality in bacteria necessitates the analysis of transcription systems that natively lack Rho and Gre factors, *e.g.*, those of Cyanobacteria.

## Materials and methods

### Reagents and oligonucleotides
DNA and RNA oligonucleotides were purchased from IBA Biotech (Göttingen, Germany) and Fidelity Systems (Gaithersburg, MD, USA). DNA oligonucleotides and RNA primers are listed in *Table 2*. NTPs were from Jena Bioscience (Jena, Germany). Tagetitoxin (TGT) was from Epicentre (Madison, WI, USA), 8-methoxypsoralen (8-MP) was from Sigma (St. Louis, MO, USA). The following buffers were used for the TEC assembly and transcription assays: TB0 (40 mM HEPES-KOH pH 7.5, 80 mM KCl, 5% glycerol, 0.1 mM EDTA, and 0.1 mM DTT), TB1 (TB0 supplemented with 1 mM $MgCl_2$), TB2 (TB0 supplemented with 2 mM $MgCl_2$ and 300 mM KCl) and TB10 (TB0 supplemented with 10 mM $MgCl_2$).

### Proteins
All proteins were expressed in *E. coli* Xjb(DE3) (Zymo Research, Irvine, CA). The wild-type, ΔLL (β'Δ P251-S263→GG), ΔRL (β'ΔN309-K325) and ΔGL (βΔR368-P376→GG) RNAPs were purified by Ni-, heparin and Q-sepharose chromatography as described previously (*Svetlov and Artsimovitch, 2015*). *E. coli* NusG was captured from the lysate by Ni-sepharose, the N-terminal hexa-histidine tag was cleaved by TEV-protease, imidazole was removed by dialysis, and the un-cleaved NusG, the cleaved tag and the TEV-protease were absorbed by passing the NusG solution over the Ni-sepharose. *E. coli* GreA containing C-terminal hexa-histidine tag was captured from lysate by Ni-sepharose followed by gel filtration as described (*Perederina et al., 2006*). All proteins were dialyzed against the storage buffer (50% glycerol, 20 mM Tris-HCl pH 7.9, 150 mM NaCl (1M NaCl for GreA), 0.1 mM EDTA, 0.1 mM DTT) and stored at −20℃. Plasmids are listed in *Table 3*. Sequences of the plasmids are provided as *Supplementary file 5* (plasmids.fas).

**Table 2.** DNA oligonucleotides and RNA primers used in this study.

| Name | type | | Sequence (5'→3') | Employment |
|------|------|------|----------|------------|
| S041M | tDNA | | GCTACTCTACTGACATGATGCCTCCTCT**X**GAACCTTAGATCGCTACAAGT | *Figures 2,6–7* |
| S154S | tDNA | | GCTACTCTACTGACATGATGCCTCCTCTG**S**AACCTTAGATCGCTACAAGT | *Figure 5—figure supplement 1* |
| S155S | tDNA | | GCTACTCTACTGACATGATGCCTCCTCT**S**GAACCTTAGATCGCTACAAGT | *Figure 5* |
| S042 | ntNA | | ACTTGTAGCGATCTAAGGTTCCAGAGGAGGCATCATGTCAGTAGAGTAGC | *Figures 2,5–7* |
| S150 | ntDNA | | ACTTGTAGCGATCTAAGGTT**A**CAGAGGAGGCATCATGTCAGTAGAGTAGC | *Figure 7A* |
| S056M | tDNA | | GCTACTCTACTGCAATGACGTCTCCTCT**X**GAACCTTAGATCGCTACAAGT | *Figures 2C, 3B, 7A* |
| S076 | tDNA | | GCTACTCTACTGCAATGACGTCTCCTCTGGAACCTTAGATCGCTACAAGT | *Figure 3* |
| S057 | ntDNA | | ACTTGTAGCGATCTAAGGTTCCAGAGGAGACGTCATTGCAGTAGAGTAGC | *Figures 2C,3,7A* |
| S152 | ntDNA | | ACTTGTAGCGATCTAAGGTTC**G**AGAGGAGACGTCATTGCAGTAGAGTAGC | *Figures 3C,7A* |
| S153 | ntDNA | | ACTTGTAGCGATCTAAGGTT**GG**AGAGGAGACGTCATTGCAGTAGAGTAGC | *Figure 3C* |
| S173 | ntDNA | | ACTTGTAGCGATCTAAGGTT**AA**AGAGGAGACGTCATTGCAGTAGAGTAGC | *Figure 3—figure supplement 3* |
| S114 | tDNA | | CGTACTCTACTCGAATAGCATCTCCTCTGGAACCTTAGATCGTCACAAGT | *Figure 3C* |
| S115 | ntDNA | | ACTTGTGACGATCTAAGGTTCCAGAGGAGATGCTATTCGAGTAGAGTACG | *Figure 3C* |
| S170 | tDNA | Atto680- | TGGTGTCTGCTGTCCGTCTGCCTCCTCTGTAGTCTGTGCTCGTGTCTGGT | *Figures 4,7A* |
| S171 | ntDNA | | ACCAGACACGAGCACAGACTACAGAGGAGGCAGACGGACAGCAGACACCA | *Figures 4,7A* |
| S224 | ntDNA | | ACCAGACACGAGCACAGACTA**A**AGAGGAGGCAGACGGACAGCAGACACCA | *Figure 7A* |
| R024 | RNA | Atto680- | CUCACAACCAGAGGAG | *Figures 2,5–7* |
| R052 | RNA | Atto680- | CUCACAACCAGAGGAG**Y**C | *Figure 3* |
| R079 | RNA | Atto680- | CAACACAACAGAGGAG | *Figures 4,7, Figure 5—figure supplement 1* |

X = 6-methyl-isoxanthopterin; S = 6-thioguanine; Y = 2-aminopurine

Mismatches in ntDNA are marked in blue and underlined.

R079 is a chimeric oligo: six 5' nucleotides are DNA.

## TEC assembly

TECs (1 µM) were assembled by a procedure developed by Komissarova *et al* (***Komissarova et al., 2003***). An RNA primer was annealed to the template DNA, incubated with RNAP for 10 min, and with the non-template DNA for 20 min at 25°C. RNA, template DNA, non-template DNA and RNAP were present at 1–2 µM during assembly. The exact ratios between the TEC components in different assays are listed in *Table 4*. The assembly was carried out in TB0 buffer for TECs used in backtracking, RNA cleavage and dinucleotide release experiment or in TB10 buffer for TECs used in all the other experiments. In the nucleotide addition experiments the assembled TEC16 were used. In the pyrophosphorolysis and NusG binding experiments the assembled TEC16s were pre-extended into TEC17s with 5 µM ATP or GTP, respectively. In the former case, the excess of ATP was further

**Table 3.** *E. coli* protein expression vectors used in this study.

| Name | Description | Source/reference |
|------|-------------|------------------|
| pVS10 | wild-type RNAP (T7p-α-β-β'_His$_6$-T7p-ω) | (***Belogurov et al., 2007***) |
| pTG011 | ΔβGL RNAP (T7p-α-His$_6$_β[ΔR368-P376→GG]-β'-ω) | this work |
| pMT041 | Δβ'RL RNAP (T7p-α-β-β'[ΔN309-K325]_TEV_His$_{10}$-T7p-ω) | this work |
| pHM001 | Δβ'LL RNAP (T7p-α-β-β'[ΔP251-S263→GG]_TEV_His$_{10}$-T7p-ω) | this work |
| pIA578 | GreA (T7p-GreA_His$_6$) | (***Perederina et al., 2006***) |
| pGB043 | NusG (T7p-His$_6$_TEV_NusG) | made by GB in Artsimovitch lab. |

Sequences of the plasmids are provided as **Supplementary file 5** (plasmids.fas).

removed by passing through the desalting spin columns (40K cutoff) with TB10 buffer. In the RNA cleavage experiments TEC18s were directly assembled in TB0 buffer.

## Nucleotide addition and RNA cleavage measurements

Time-resolved measurements of nucleotide addition were performed in an RQF 3 quench-flow instrument (KinTek Corporation, Austin, TX). The reaction was initiated by rapid mixing of 14 µl of 0.4 µM TEC with 14 µl of 400 µM NTP. Both TEC and NTP solutions were prepared in TB10 buffer. The reaction was allowed to proceed for 0.004–10 s at 25°C, quenched with 86 µl of 0.5 M HCl and immediately neutralized by adding 171 µl of neutralizing-loading buffer (94% formamide, 290 mM Tris base, 13 mM $Li_4$-EDTA, 0.2% Orange G). RNA extension was also followed in 6-MI fluorescence assays by withdrawing 8 µl aliquots from the fluorometer cuvette into 12 µl of gel loading buffer (94% formamide, 20 mM $Li_4$-EDTA and 0.2% Orange G). RNA cleavage was monitored by manual mixing of 50 µl of 0.2 µM TEC in TB0 buffer with 50 µl of 16 µM GreA in TB2 buffer. The aliquots (8 µl) were withdrawn at the indicated time points and quenched with 12 µl of the gel loading buffer. The TEC solutions were supplemented with 4 µM NusG where indicated. RNAs were separated on 16% denaturing polyacrylamide gels and visualized with Odyssey Infrared Imager (Li-Cor Biosciences, Lincoln, NE); band intensities were quantified using ImageJ software (*Abramoff et al., 2004*).

## Time resolved measurements with the 6-MI fluorescent beacon

Measurements were performed in an Applied Photophysics (Leatherhead, UK) SX.18 MV stopped-flow instrument at 25°C. 6-MI fluorophore was excited at 340 nm and emitted light was collected through 400 nm longpass filter. At least three individual traces were averaged for each reported curve. The nucleotide addition, pyrophosphorolysis and RNA cleavage reactions were initiated by mixing 60 µl of 0.2 µM TEC with 60 µl of 400 µM NTP, 1000 µM $PP_i$ and 16 (or 4) µM of GreA, respectively. TEC solutions were supplemented with 4 µM NusG where indicated. The NusG binding reaction was initiated by mixing 60 µl of 0.4 µM TEC with 60 µl of 0.2–20 µM NusG. In the nucleotide addition and pyrophosphorolysis experiments reactant solutions were prepared in TB10 buffer, whereas in NusG binding experiments TB1 buffer was used. In the RNA cleavage experiments, TEC and GreA solutions were prepared in TB0 and TB2 buffers, respectively.

## Time resolved measurements of the dinucleotide release

Measurements were performed in an Applied Photophysics SX.18 MV stopped-flow instrument at 25°C. 2-AP fluorophore was excited at 320 nm and emitted light was collected through 375 nm longpass filter. At least three individual traces were averaged for each reported curve. The RNA cleavage reactions were initiated by mixing 60 µl of 0.2 µM TEC with 60 µl of 16 (or 4) µM of GreA. TEC and

**Table 4.** TEC assembly ratios and reaction buffers.

| | Concentrations during assembly, µM | | | | Assembly buffer | Reaction buffer* | | | |
|---|---|---|---|---|---|---|---|---|---|
| | RNA | tDNA | ntDNA | RNAP | | TEC | additive | | |
| Nucleotide addition (gel) | 1 | 1.4 | 2 | 1.5 | TB10 | TB10 | TB10 | | *Figure 2B* |
| RNA cleavage (gel) | 1 | 1.4 | 2 | 1.5 | TB0 | TB0 | TB2 | | *Figure 3B* |
| Forward translocation (nucleotide addition) | 1.4 | 1 | 2 | 1.5 | TB10 | TB10 | TB10 | | *Figure 2B* |
| Backward translocation (pyrophosphorolysis) | 1.4 | 1 | 2 | 1.5 | TB10 | TB10 | TB10 | | *Figure 2C* |
| Backtracking (RNA cleavage) | 1.4 | 1 | 2 | 1.5 | TB0 | TB0 | TB2 | | *Figure 3B* |
| NusG binding | 1.4 | 1 | 2 | 1 | TB1 | TB1 | TB1 | | *Figure 6C* |
| Equilibrium 6-MI assays | 1.4 | 1 | 2 | 1.5 | TB10 | TB10 | | | *Figures 2D,6–7* |
| Dinucleotide release (RNA cleavage) | 1 | 1.4 | 2 | 1.5 | TB0 | TB0 | TB2 | | *Figure 3* |
| 8-MP crosslinking | 1.2 | 1 | 1 | 1.5 | TB10 | TB10 | | | *Figures 4,7A* |
| 6-TG crosslinking | 1 | 1 | 2 | 1.5 | TB10 | TB10 | | | *Figure 5* |

* In time resolved assays the equal volumes of the TEC and the additive solutions were mixed to initiate the reaction.

**Table 5.** The number of repeats for each experiment.

| Figure | Data | Number of experiments | | | | | |
| | | with independently assembled TECs | | including the experiments with the same TEC preparation | | with independently assembled TECs not in the figures | |
| | | control | +NusG | control | +NusG | control | +NusG |
|---|---|---|---|---|---|---|---|
| 2B | WT catalysis | 4 | 3 | 8 | 6 | | |
| | WT translocation | 3 | 2 | >12 | >8 | | |
| 2C | WT pyrophosphorolysis | 2 | 2 | >8 | >8 | | |
| 2D | WT TGT binding | 2 | 2 | | | | |
| 3BC | WT RNA cleavage | 2 | 2 | 3 | 4 | 6 | 2 |
| | WT 6-MI | 2 | 2 | >8 | >8 | | |
| | WT 2-AP | 2 | 2 | >8 | >8 | | |
| | ΔRL 2-AP | 2 | 2 | >8 | >8 | | |
| | ΔLL 2-AP | 2 | 2 | >8 | >8 | | |
| | ΔGL 2-AP | 2 | 2 | >8 | >8 | | |
| | WT mm1 2-AP | 2 | 2 | >8 | >8 | | |
| | WT mm1-2 2-AP | 2 | 2 | >8 | >8 | | |
| | WT 3'mm 2-AP | 2 | 2 | >8 | >8 | | |
| 3S3 | WT 6-MI | 2 | 2 | 5 | 5 | 1 | 1 |
| | WT 2-AP | 1 | 1 | 3 | 3 | | |
| | WT mm1-2 AA 2-AP | 1 | 1 | 7 | 8 | | |
| | WT mm1-2 GG 2-AP | 2 | 2 | >8 | >8 | | |

The experiments reported in the figures were performed with the same batch of GreA. The older and the newer experiments cannot be directly combined with the reported experiments due to the variations in the specific activity of the GreA preparations. However, the relative effect of NusG on the reactions involving backtracking can be estimated from all available data. In the WT TEC NusG inhibits reactions that involve backtracking:
2.61 ± 0.28 fold (n = 5, 2 µM GreA);
2.62 ± 0.22 fold (n = 6, 8 µM GreA)

| Figure | Data | control | +NusG | control | +NusG | control | +NusG |
|---|---|---|---|---|---|---|---|
| 4AB | WT 8-MP | 7 | 3 | | | | |
| | ΔRL 8-MP | 3 | 2 | | | | |
| | ΔLL 8-MP | 3 | 2 | | | | |
| | WT TEC18+TGT 8-MP | 2 | | | | | |
| 5AB 5S2 | WT 6-TG | 7 | 3 | | | | |
| | ΔRL 6-TG | 2 | 2 | | | | |
| | ΔLL 6-TG | 3 | 2 | | | | |
| | WT TEC18+TGT 6-TG | 2 | | | | | |
| 6AB,C(*Left*) | WT 6-MI | 2 | 2 | >5 | >3 | >7 | >7 |
| | ΔRL 6-MI | 2 | 2 | 5* | 3* | | |
| | ΔLL 6-MI | 2 | 2 | 5* | >3 | | |

\* Except TEC19.
SD of the fluorescence measurements with the same TEC preparation = 2.3 ± 1.3% (n = 122). SD of the fluorescence measurements with the independently assembled TECs = 16% (n = 8, WT TEC17 measured with different batches of the fluorescent oligonucleotides). Accordingly, the primary data from all fluorescent experiments cannot be directly combined with a figure but some NusG effects can be estimated with the highest accuracy and precision from all available data. NusG effects on the fluorescence intensity of the WT TECs:
(TEC17NusG–TEC16NusG)/(TEC17-TEC16) = 1.45 ± 0.09 (n = 9)
(TEC18NusG–TEC16NusG)/(TEC18-TEC16) = 0.99 ± 0.02 (n = 7)
(TEC19NusG–TEC16NusG)/(TEC19-TEC16) = 0.50 ± 0.35 (n=3)

| Figure | Data | control | +NusG | control | +NusG | control | +NusG |
|---|---|---|---|---|---|---|---|
| 6C (*Right*) | WT+NusG 6-MI | 2 | | >6 | | | |
| 7A | TEC16-19 6-MI | 2 | | 9 (except TEC19) | | | |
| | TEC16-19 6-MI | 2 | | 4 (except TEC19) | | | |
| | TEC16-18 8-MP | 2 | | 2 | | | |

GreA solutions were prepared in TB0 and TB2 buffers, respectively. TEC solutions were supplemented with 4 µM NusG where indicated.

## Equilibrium measurements with the 6-MI fluorescent beacon

Equilibrium levels of fluorescence were determined by continuously recording light emission at 420 nm (excitation at 340 nm) with an LS-55 spectrofluorometer (PerkinElmer, Waltham, MA) in a 16.160-F/Q/10 quartz cuvette (Starna) at 25°C. The assembled TECs were diluted at 100 nM into 200 µl of TB10 buffer and the NTP substrates (5 µM) and/or the increasing concentrations of TGT (where indicated) were sequentially added into the cuvette. The reaction was allowed to proceed for up to two minutes between each successive addition to ensure that the fluorescence reached the equilibrium level.

## 8-methoxypsoralen (8-MP) and 6-thioguanine (6-TG) crosslinking

In 8-MP (mono-adduct absorption maximum 342 nm [*Tessman et al., 1985*]) crosslinking experiments the reaction mixture contained 1 µM TEC, 0.92 mM 8-MP, 6.3% DMSO in TB10 buffer. In 6-TG (absorption maximum 340 nm [*Karran and Attard, 2008*]) crosslinking experiments the reaction mixture contained 1 µM TEC in TB10 buffer. NTPs (5 µM), TGT (5 µM) and NusG (2.5 µM) were added were indicated. TEC samples (5 µl) were placed in an 18-well circular tray (Ø=26 mm, all wells equidistant from the center) in a closed thermally controlled chamber with the UV LED (P8D1 365 nm, Seoul Viosys, Ansan, Korea) in the top center (height=17 mm). Samples were exposed to UV for 30 min at 25°C, 4 µl aliquots were quenched with 6 µl of loading buffer and separated on 14% denaturing PAGE gel. ATTO680 labeled species were visualized with Odyssey Infrared Imager (Li-Cor Biosciences, Lincoln, NE); band intensities were quantified using ImageJ software (*Abramoff et al., 2004*).

## TEC-NusG model

The composite model was generated using the structure of *T. thermophilus* TEC with the NTP analogue (*Vassylyev et al., 2007*) (PDB ID 2O5J, the lineage specific domain (β′132–456) omitted), NusG NTD from the model of *T. thermophilus* NusG-RNAP complex in *Martinez-Rucobo et al. (2011)*, NusG CTD (G187-I248) from the crystal structure of *Aquifex aeolicus* NusG (*Steiner et al., 2002*) (PDB ID 1M1G) and αCTDs from the crystal structure of *E. coli* holoenzyme (*Murakami, 2013*) (PDB ID 4YG2). The duplex DNA immediately upstream of the RNA:DNA hybrid was modeled de novo as described in results; the downstream DNA outside of the TEC was extended with the canonical DNA duplex. The positions of NusG CTD and αCTD were chosen arbitrary within the volume permitted by the length of the flexible linkers tethering those domains to the TEC. Parts of the linkers were modeled de novo using ModLoop RRID:SCR_008395 (*Fiser and Sali, 2003*). NusG CTD and αCTDs are highly conserved in bacteria but are likely irrelevant to the NusG effects in the present study. The above considerations justify the use of heterologous NusG CTD and αCTD in the composite model solely for the illustrative purposes. The model geometry was evaluated using Mol-Probity RRID:SCR_014226 (*Chen et al., 2010*). The atomic coordinates of the TEC-NusG complex are provided as *Figure 8—source data 1* (NusG-TEC.pdb). To generate *Figures 1* and *8A* the simplified surfaces (Gaussian resolution 6, B-factor 50) were calculated and rendered in PyMOL Molecular Graphics System, RRID:SCR_000305, (Schrödinger, New York, NY), exported in VRML format, converted to OBJ format using Meshlab and further simplified using sculpting tools of Meshmixer (Autodesk Inc. San Rafael, CA). The resulting meshes were imported into and rendered in Rhinoceros 4.0 RRID:SCR_014339 (Robert McNeel & Associates, Seattle, WA). The superimposition of *Aquifex aeolicus* NusG NTD (PDB ID 1M1G, residues A9-50, A133-185) and *Pyrococcus furiosus* Spt5 NTD (PDB ID 3P8B, residues B4-82) was performed using COLORBYRMSD PyMOL plugin (by S. Shandilya, J. Vertrees, T. Holder).

## Data analyses

Time-resolved nucleotide incorporation and the forward translocation data were simultaneously fit to a three-step model using the numerical integration capabilities of KinTek Explorer software (*Johnson, 2009*) (KinTek Corporation, Austin, TX). The model postulated that the initial TEC16 slowly and reversibly interconverts between inactive and active states and, upon the addition of the NTP substrate, undergoes an irreversible transition to TEC17, followed by irreversible translocation (*Supplementary file 1*) (*Malinen et al., 2014*). Pyrophosphorolysis, backtracking, RNA cleavage and dinucleotide release data were fit to the stretched exponential function and the median reaction

times were used in place of half-lives to quantify the reaction progress (*Supplementary file 2*). Equilibrium titration data were fit to the dissociation equilibrium equations that accounted for changes in concentrations of all reactants upon complex formation using Scientist 2.01 software (Micromath, Saint Louis, MO) (*Supplementary file 3*). NusG binding data were fit to a one-step reversible binding model (*Supplementary file 4*). Numerical values of the reaction rate constants and the median reaction times are presented in *Table 1*. The number of repeats for each experiment is indicated in *Table 5*.

## Acknowledgements

The authors would like to thank Irina Artsimovitch for critically reading the manuscript and providing expression plasmids; Thadée Grocholski, Henri Malmi, Salli Keinänen and Pavlína Gregorova for assistance with protein purification, cloning and biochemical experiments; Anssi M Malinen for helpful discussions and Jani Sointusalo for making the crosslinking device. Essential equipment was contributed by Walter and Lisi Wahl Foundation.

## Additional information

### Funding

| Funder | Grant reference number | Author |
|--------|------------------------|--------|
| Suomen Akatemia | grant #286205 | Matti Turtola<br>Georgiy A Belogurov |
| Turun Yliopisto | Graduate Student Fellowship | Matti Turtola |

The funders had no role in study design, data collection and interpretation, or the decision to submit the work for publication.

### Author contributions

MT, Designed and performed the biochemical experiments, Interpreted the results, Wrote the manuscript; GAB, Supervised the study, Performed structural modeling, Interpreted the results, Wrote the manuscript, Conception and design

### Author ORCIDs

Matti Turtola, http://orcid.org/0000-0003-1694-1027
Georgiy A Belogurov, http://orcid.org/0000-0002-3070-6843

## Additional files

### Supplementary files

• Supplementary file 1. Model employed for fitting of nucleotide addition and translocation data.

• Supplementary file 2. Model employed for fitting pyrophosphorolysis, backtracking, RNA cleavage and dinucleotide release data.

• Supplementary file 3. Model employed for fitting equilibrium titration with TGT.

• Supplementary file 4. Model employed for fitting NusG binding kinetics.

• Supplementary file 5. Plasmid sequences.

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
