## [Decision Letter]

Thank you for submitting your article "NusG inhibits RNA polymerase backtracking by stabilizing the minimal transcription bubble" for consideration by *eLife*. Your article has been reviewed by three peer reviewers, and the evaluation has been overseen by a Reviewing Editor and Kevin Struhl as the Senior Editor. The reviewers have opted to remain anonymous.

The reviewers have discussed the reviews with one another and the Reviewing Editor has drafted this decision to help you prepare a revised submission.

Summary:

Overall the reviewers find that the manuscript reports a significant advance in the understanding of the interactions of DNA and RNA at the upstream edge of the transcription bubble in elongating transcription complexes and in the way that NusG, the only universally conserved transcription elongation factor, affects these interactions in vitro in the *E. coli* model system. The authors report results from elegant enzymatic, fluorescence, and crosslinking assays to establish the following main points.

1) The -10 DNA base-pair in the translocated elongation complex formed after a round nucleotide addition and before binding of the next NTP substrate is partially paired but not stacked on the upstream DNA duplex.

2) NusG inhibits backtracking of RNA and DNA through the elongation complex without a major effect on interconversion of the pretranslocated and translocated states.

3) The antibacktracking effect of NusG results from NusG promotion of DNA:DNA annealing at the upstream fork-junction of the transcription bubble, possibly due to a conserved loop in the NusG NTD that may inhibit fraying of the upstream fork-junction.

4) The RNA polymerase lid loop, but not the rudder or gate loop, contributes the annealing/melting dynamics of the upstream fork-junction and thereby affects backtracking.

Essential revisions:

Although these findings are of a strong biochemical significance, a major concern of multiple reviewers is that the extent to which backtracking inhibition by NusG contributes to gene expression or regulation in vivo is not established. The impact of the findings is significantly lessened if this effect is observable in vitro but does not come into play in vivo in any significant way. Accordingly, for the manuscript to be suitable for high-profile publication, the authors must provide either experimental evidence for the in vivo significance of their findings or a concise description in the discussion of how already published findings establish the in vivo significance of their results.

In addition to this major concern, the authors must address the following issues raised by the reviewers.

1) Both reviewer #1 and reviewer #3 raised concerns about the conclusion that NusG does not affect the stabilities or intercconversion of the pre- and post-translocated states of an elongation complex. These concerns must be addressed and the authors should modify their conclusions accordingly.

2) Both reviewer #1 and reviewer #3 raised concerned about overly categorical statements in the Introduction. Please revise the Introduction to address these concerns.

3) Reviewer #3 has made suggestions to improve the impact of your manuscript by more completely illustrating or explaining your model and some of its implications. Comments #2 and #3 from reviewer #3 should be addressed in a revision.

4) Both reviewer #2 and reviewer #3 raised concerns about the way error analysis is presented in some of the figures. These concerns must be addressed.

These major issues as well as many additional minor points and suggestions for revisions are included in the detailed critiques supplied by the reviewers that are appended below. Although some of these points are straightforward and easily fixed in the manuscript, please note that there were no significant disagreements among the reviewers about these critiques. Every effort should be made to address all these other comments in a revision or to explain why no revision is needed.

*Reviewer #1:*

This is a very nice paper from the Belugurov lab concerning the molecular mechanism of the NusG elongation factor in *E. coli*. This factor is extensively described in the literature and has several seemingly independent activities; in addition NusG from different bacterial species has quite opposite effects on distinct RNAP activities. The authors provide rather convincing biochemical and -physical evidence that NusG increases processivity by inhibiting backtracking of RNAP by stabilising the transcription bubble (using in vitro transcription and chemical crosslinking experiments, and fluorescence measurements). This mechanism is in good agreement with the structural understanding of the elongation complex and not unexpected as such. A slight shortcoming of the work is the narrow range of techniques applied, and the absence of any data supporting the in vivo relevance of the suggested mechanism. In other words, does it make any difference to RNAP elongation and gene expression whether NusG modulates the reannealing of the DNA strands at the upstream edge of the transcription bubble? In the cell the elongation complex includes additional factors, NusA and even rho – how would these alter the effect of NusG on RNAP? Its very interesting to study and describe NusG mechanisms by measuring the incorporation of single nucleotides and recombinant factors in vitro, but does e.g. the genome-wide occupancy of RNAP change in strains harbouring NusG mutants which disrupt the predicted mechanism? I feel that these kind of data would elevate the impact of the type of results presented in the manuscript to be of interest of the general audience of *eLife*. Alternatively, providing structural information of the RNAP-NusG elongation complex (e.g. using cryoEM) would propel the structural hypothesis beyond a model (Figure 7, and subsection “Mapping the structure of the upstream fork junction”) and provide an additional perspective that would raise the impact of the work required for *eLife*. Having said that, the data are of very high quality and the conclusions are generally not over interpreted, and the results are of interest to the transcription community, in particular researchers that are working on the detailed molecular mechanisms of RNAP. I enjoyed reading this paper.

*Reviewer #2:*

This manuscript presents a wide range of related biochemical and biophysical experiments that present a reasonably self-consistent picture of structure and interactions at the upstream edge of the elongation complex in bacterial RNA polymerase, specifically focusing the relative effects on those measurements of the elongation factor NusG. While any one assay in this toolbox has a range of caveats with respect to the precision of its conclusions, that very different assays present a similar picture provides compelling support for the derived model. The authors are appropriately cautious for the most part, in their broad analyses. For example, fluorescent base analogs are sensitive to more than just base pairing (indeed, they are reflective of pairing-related changes in stacking) and cross linking propensities are are not direct measures of distance.

A bigger picture critique might be that these experiments are all carried out on scaffold complexes and so are 'artificial.' Again, the relative consistency of the results assuages such concerns and the power that this approach uniquely yields cannot be ignored. Most time courses are consistent with biologically relevant time scales.

In general, the data analysis in this manuscript is thoughtful and careful. However, in Figure 6, the concentration dependence of kinetic profiles is used to extract forward and reverse rate constants, which are then used to derive binding constants. While the data fit reasonably well, there are some systematic deviations at the extremes. Perhaps more importantly, interdependence of the two fit parameters should be discussed, as they are then used together to calculate Kd and its range. The authors should consider a somewhat more detailed analysis of this fit in the supplementary material. At the very least, they should indicate the equations fit.

*Reviewer #3:*

Turtola and Belogurov report a tour de force of biochemical assays to probe the effect of NusG on the transcription elongation complex. Because NusG (with its ortholog Spt5) is the only universally conserved transcription elongation factor across all kingdoms of life and because NusG/Spt5 is involved in the regulation of transcript elongation known to affect the expression of many genes, understanding the action mechanism of NusG/Spt5 is among the more important extant questions in the field of transcriptional regulation. Overall, Turtola and Belogurov provide compelling evidence that NusG from the model bacterium *E. coli* inhibits backtracking of elongation complexes by favoring pairing of the upstream edge of the DNA bubble. The DNA bubble is integral to the structure and activity of transcription complexes. The authors also provide new insight into the exact configuration of this upstream "fork-junction" by showing compelling evidence that the -10 DNA bp immediately upstream of the RNA:DNA hybrid (counting from the template nt positioned in the NTP binding site) is partially re-paired, but stacked on neither the RNA:DNA hybrid nor the exiting upstream duplex, whereas the -11 DNA bp is fully re-paired into an apparently canonical bp stacked on the exiting DNA duplex. The authors provide a hypothetical structural model that positions a conserved loop of NusG in an appropriate location to aid upstream fork-junction reannealing and that may explain the partially complementary role of the RNA polymerase lid-loop in the reannealing process. Although the authors findings do not explain the entire and surprisingly wide range of effects NusG can have on elongation complexes in different organisms, they provide an important framework for future studies. The work is likely to be a high interest the researchers studying the regulation of transcription elongation, which is an area currently undergoing significant growth because so many metazoan genes turn out to be regulated during the process of RNA synthesis rather than at steps preceding transcription initiation. There are a couple significant points that the authors will need to address before the manuscript would be acceptable for publication. These major points, along with a number of minor points are enumerated below.

1) I don't buy the argument that the authors have shown that NusG has no effect on steps other than backtracking, in particular on the rate of pause-free transcription elongation. This claim by the authors will need to be revised for the following reasons. First, a previous study that was much more sensitive to small effects that could aggregate across many successive rounds of nucleotide addition reached the opposite conclusion (Herbert et al., 2010), yet the authors do not explain how the present results invalidate these previous results and conclusion. Indeed, their own result shows a modest increase in the overall rate of a single nucleotide turnover reaction even though it is within their experimental error (Figure 2; see bar graphs). Put another way, the modest effects of NusG on pause-free elongation reported previously are not detectable within the experimental error of the present measurements. Further, the authors have looked at only a single template position, whereas the contribution of translocation to elongation and the effect of sequences on upstream fork-junction reannealing and its contribution to translocation and to elongation are certain to vary at different template positions. At least for the basic contribution of translocation to elongation rate, these effects are demonstrated to vary among template positions (e.g., Dangkulwanich et al., 2013 *eLife* 2: e00971). Given that the authors have provided no evidence to rule out the small effects of NusG previously shown, the authors should modify their conclusion to state that NusG has at most a modest effect on pause-free transcription rate and a greater effect on backtracking, a conclusion indeed essentially the same as that reached by Herbert et al.

2) The authors can improve their manuscript by slightly expanding the presentation and discussion of their model for NusG interaction with the upstream fork junction. The authors have appropriately presented this model conservatively given the lack of direct evidence for the NusG-DNA interaction they propose. That said, the impact of the manuscript can be significantly improved by more clearly presenting the model and its implications, as described here and in the next point. First, the authors can much better illustrate their proposed structure for the upstream fork junction. Figure 7 is OK as far as it goes, but it does not aid the viewer in understanding the fork-junction structure nor does it identify the 5 aa loop in NusG proposed to aid in junction reannealing. I suggest the authors prepare additional panels that show two things. In one additional panel, they should show the fork junction from a different perspective so that the partially paired nature of the -10 by and stacking of the -11 bp is more easily understood. Why not use actual base structures for this panel, rather than the simplified stick diagrams that don't really illustrate stacking (or the absence of stacking)? In a second additional panel, the authors should show the sequence of the 5 aa loop and parts of the flanking sheet and helix from *E. coli* aligned with small number of examples from a diverse evolutionary range of organisms. Please include numbering in this panel so that readers can readily relate the sequences shown to the sequences to full-length NusG/Spt5 sequences.

3) A key question about the authors conclusion is why NusG can inhibit backtracking while having so much less effect on interconversion of the pretranslocated and translocated registers (this discrepancy is equally puzzling whether the effect on the interconversion is modest, as the preponderance of evidence would suggest as described in point 1, or nonexistent, as the authors claim). The authors suggest an entirely reasonable hypothesis that hybrid translocation and downstream DNA translocation may be two distinct, if interconnected, steps, rather than a single translocation event, and that NusG will principally affect events, like backtracking, for which hybrid translocation is rate-limiting). I think this is an important idea, and that the authors could expand their description of it somewhat without crossing the line to overinterpretation of their results. From my perspective, it would be appropriate to say that not only can the preferential effect of NusG on backtracking potentially be explained by the tw0-translocation model, but also that the difficulty of reconciling the large effect of NusG on backtracking and the modest effect on translocation and elongation more generally in the context of a single-step translocation model supports the a two-step translocation mechanism. As long as it's stated this way, and not that the results somehow prove the two-step model, then I think the authors could improve the impact of their manuscript by slightly expanding the discussion of this point.

---

## [Author Response]

*[…] Essential revisions:*

*Although these findings are of a strong biochemical significance, a major concern of multiple reviewers is that the extent to which backtracking inhibition by NusG contributes to gene expression or regulation in vivo is not established. The impact of the findings is significantly lessened if this effect is observable in vitro but does not come into play in vivo in any significant way. Accordingly, for the manuscript to be suitable for high-profile publication, the authors must provide either experimental evidence for the in vivo significance of their findings or a concise description in the discussion of how already published findings establish the in vivo significance of their results.*

We have added a section into the Discussion where we contrast the universally conserved stimulatory effect of NusG family proteins on transcription elongation with the role of NusG in promoting Rho-dependent termination, the major functionality of NusG in bacteria. We suggest that the universal conservation of the stimulatory effect of NusG NTD is the major argument underscoring the significance of this activity in vivo. We further suggest that NusG-Rho cooperation masks other effects of NusG in bacteria and refer to the examples from the eukaryotic systems where the stimulatory activity of SPT4/5 has been linked to phenotypes.

*In addition to this major concern, the authors must address the following issues raised by the reviewers.*

*1) Both reviewer #1 and reviewer #3 raised concerns about the conclusion that NusG does not affect the stabilities or intercconversion of the pre- and post-translocated states of an elongation complex. These concerns must be addressed and the authors should modify their conclusions accordingly.*

The reviewers’ concerns are of a different nature. Reviewer #1 doubts the inferences from our experiments. Reviewer #3 questions whether some inferences from our experiments could be generalized as broadly as we did, in light of the existing conflicting evidence. Accordingly, we answer the reviewers’ concerns individually.

(i) Reviewer #1 raised concern about our conclusions drawn from the TGT titration experiments. We now explain in detail how we infer the effect of NusG on the translocation bias from the TGT titration experiments in Figure 2—figure supplement 4. We also changed the sentence describing the TGT effects in the main text. It now reads as follows:

“TGT was equally potent in biasing the TEC towards the pre-translocated state in the presence and absence of NusG, suggesting that NusG does not affect the equilibrium between the post- and pre-translocated states.”

We think that the original sentence might have been confusing because it described the measurement of the affinity of the TEC for TGT instead of emphasizing that the interconversion of the post-translocated TEC into the pre-translocated state takes place upon increasing TGT concentration.

We further emphasize that TGT experiment is not the only evidence that we use to infer the absence of NusG effects on the backward translocation and the equilibrium between the pre- and the post-translocated states. We measured the effect of two agents, PP_i_ and TGT, which bias TEC backward in the presence and absence of NusG. The field largely agrees that TGT is a transcription inhibitor that binds in the RNAP active site and stabilizes the pre-translocated state (Artsimovitch et al., 2011; Malinen et al., 2012; Yuzenkova et al., 2013). We have previously suggested that TGT is in fact a high affinity analogue of PP_i_ (Malinen et al., 2012). The advantage of using TGT over PP_i_ is that no chemical interconversion takes place and the system can be investigated in an equilibrium setup.

Pyrophosphorolysis is the reversal of the nucleotide addition and is catalyzed by the pre-translocated TECs. NusG had no effect on the rate of pyrophosphorolysis reaction (Figure 2) by initially post-translocated TEC, suggesting that NusG has no effect on the backward translocation rate. Pyrophosphorolysis experiment also indirectly suggests that NusG has no effect on the equilibrium between the pre- and the post-translocated states, because NusG has no effect on the forward translocation rate in a separate experiment (Figure 2).

Similarly, the lack of NusG effect on the potency of TGT to bias the initially post-translocated TEC into the pre-translocated state (Figure 2) suggests that NusG has no effect on the equilibrium constant between the pre- and post- translocation states (Figure 2—figure supplement 4). TGT experiment also indirectly suggests that NusG has no effect on the backward translocation rate, because NusG has no effect on the forward translocation rate in a separate experiment (Figure 2).

(ii) Reviewer #2 suggested that results obtained in our system do not rule out a possibility that NusG affects the on-pathway elongation at some other sequence positions as reported by Herbert et al. 2010.

We addressed the reviewer request in the following way:

1. We added “measurably” to the sentences about the on-pathway elongation in the abstract and the significance statement. In our view, “measurably” is the concise way to state that no effect on the on-pathway elongation was observed in our experiments taking into account the experimental uncertainty.

2. We have rewritten the Discussion as follow:

“Here, we show that NusG slows backtracking but does not affect the on-pathway elongation in the non-paused TEC used in our study. In contrast, (Herbert *et al.,* 2010) concluded that NusG has a modest stimulatory effect (10-20%) on the pause-free elongation rate in addition to inhibiting backtracking. […] In any case, the effect of NusG on the pause-free elongation rate estimated by Herbert et al. is small comparing with the specific effect of NusG on the backtracking rate (~2.5 fold) that we report here.”

We find it unreasonable to soften our conclusions to the point of full congruence with the conclusions of Herbert et al. Our study agrees with Herbert et al. that NusG stimulates transcription by affecting the lateral position of the RNAP on the template DNA. However, our results suggest that NusG acts exclusively by inhibiting backtracking, whereas Herbert et al. suggest an additional marginal effect of NusG on the bias between the pre- and post-translocated states.

It is important to note that we measured the forward translocation and nucleotide addition independently. We also investigated the effects of NusG on the backward translocation under conditions where the backward translocation can be distinguished from backtracking. In contrast, the force-velocity single-molecule studies do not measure the forward translocation and nucleotide addition independently and the effects on translocation are inferred indirectly, for example by investigating the load sensitivity. The effects on the forward, backward translocation and backtracking are difficult to tell apart in a situation where backtracking is short, e.g. by only one nucleotide (Depken M, Galburt EA & Grill SW (2009) Biophys. J. 96: 2189–93).

Most importantly, Herbert et al. concluded that NusG has an additional effect on the on-pathway elongation by assuming that all backtracked TECs are paused and therefore do not contribute to the pause-free elongation rate. We argue that this assumption is only superficially plausible. The fact that many pauses are backtracked does not prove that all backtracked TECs are paused. We suggest that the long-backtracked TECs are perhaps indeed universally paused, but the TECs backtracked by one nucleotide are not necessarily paused and may display a relatively fast elongation rate. For example, backtracking occurs with the rate of ~0.3 s^-1^ in the TECs with the mismatched upstream DNA (we chose the fastest to backtrack in Figure 3). At the same time, we do not observe a measurable fraction of the backtracked state in that system before GreA is added. Considering the experimental uncertainties, we estimate that the backtracked state occupancy is ≤10%. The above two values suggest that the TEC backtracked by one nucleotide recovers with the rate ≥3 s^-1^, well within the range of pause-free elongation rates observed by Herbert et al. From the thermodynamic standpoint, the Brownian ratchet model of RNAP translocation postulates that there is the same amount of energy available for the forward translocation from the pre- to post-translocated state and from the backtracked to the pre-translocated state. Next, there is no a priori indication that the energy barrier between the 1-nt backtracked and the pre-translocated states is uniformly high to equate short backtracking with pausing.

Overall, we think that it is reasonable to retain some degree of disagreement between our conclusions and those of Herbert et al. 2010 until the rapidly elongating NusG-responsive TECs are identified and characterized individually, and it is confirmed or disproved that the 1-nt backtracked TECs are uniformly slow and do not contribute to the pause-free rate under the conditions used by Herbert et al.

*2) Both reviewer #1 and reviewer #3 raised concerned about overly categorical statements in the Introduction. Please revise the Introduction to address these concerns.*

We have altered the Introduction to address the reviewers’ concerns.

*3) Reviewer #3 has made suggestions to improve the impact of your manuscript by more completely illustrating or explaining your model and some of its implications. Comments #2 and #3 from reviewer #3 should be addressed in a revision.*

In response to comment #2 from reviewer #3 we made a multi-panel Figure 8 that illustrates the model and the structural conservation of the five amino acid loop of NusG NTD.

In response to comment #3 from reviewer #3 we modified the corresponding part of the Discussion. Specifically, we explained the concept of the sequential translocation in more detail and suggested that the specific effect of NusG on backtracking supports a two-step translocation mechanism.

*4) Both reviewer #2 and reviewer #3 raised concerns about the way error analysis is presented in some of the figures. These concerns must be addressed.*

We modified the legends of the figures and the tables. We now state where appropriate that the error bars indicate the range of duplicate measurements or SDs of several measurements (Table 5). In Table 5 we listed the number of independent replicates for each experiment.

We now also describe in Figure 6—figure supplement 2 how we accounted for the interdependence of the binding and the dissociation rate constants when determining the upper and lower bounds of the equilibrium dissociation constant of TEC-NusG complex.

*Reviewer #1:*

*This is a very nice paper from the Belugurov lab concerning the molecular mechanism of the NusG elongation factor in E. coli. This factor is extensively described in the literature and has several seemingly independent activities; in addition NusG from different bacterial species has quite opposite effects on distinct RNAP activities. The authors provide rather convincing biochemical and -physical evidence that NusG increases processivity by inhibiting backtracking of RNAP by stabilising the transcription bubble (using in vitro transcription and chemical crosslinking experiments, and fluorescence measurements). This mechanism is in good agreement with the structural understanding of the elongation complex and not unexpected as such. A slight shortcoming of the work is the narrow range of techniques applied, and the absence of any data supporting the in vivo relevance of the suggested mechanism. In other words, does it make any difference to RNAP elongation and gene expression whether NusG modulates the reannealing of the DNA strands at the upstream edge of the transcription bubble? In the cell the elongation complex includes additional factors, NusA and even rho – how would these alter the effect of NusG on RNAP? Its very interesting to study and describe NusG mechanisms by measuring the incorporation of single nucleotides and recombinant factors in vitro, but does e.g. the genome-wide occupancy of RNAP change in strains harbouring NusG mutants which disrupt the predicted mechanism? I feel that these kind of data would elevate the impact of the type of results presented in the manuscript to be of interest of the general audience of eLife. Alternatively, providing structural information of the RNAP-NusG elongation complex (e.g. using cryoEM) would propel the structural hypothesis beyond a model (Figure 7, and subsection “Mapping the structure of the upstream fork junction”) and provide an additional perspective that would raise the impact of the work required for eLife. Having said that, the data are of very high quality and the conclusions are generally not over interpreted, and the results are of interest to the transcription community, in particular researchers that are working on the detailed molecular mechanisms of RNAP. I enjoyed reading this paper.*

We have added a section into the Discussion where we acknowledge the importance of NusG-Rho cooperation in bacteria and contrast it with the universally conserved stimulatory effect of NusG family proteins on transcription elongation. We argue that the importance of NusG for regulating the activity of Rho masks the other functionalities of NusG in *E. coli*. At the same time, the anti-backtracking activity of NusG family proteins is likely determined by the fold of the NTD so it is not straightforward to construct a NusG variant with an unaltered fold that is fully devoid of such activity. We further suggest that the quantitative assessment of the in vivo role of elongation stimulation by NusG in bacteria necessitates the analysis of transcription systems that natively lack Rho, e.g., those of Cyanobacteria.

Next, we fully agree that it is reasonable to gradually increase the complexity of the in vitro system by adding more factors such as NusA, B, E, ribosome, σ, that are normally present in vivo. However, such expansion requires rigorous tests of the effects of several factors in several assays individually and in combination. In addition, many factors require considerably longer RNA for the full functionality than the one that we used to assemble TECs on the nucleic acid scaffold (16-18 nt). It is very straightforward to add NusA to our existing systems, but we do not think that the result of such *ad hoc* experiment will be conclusive. Accordingly, we suggest that the expansion of the study to include more factors is outside the scope of the present manuscript.

*Reviewer #2:*

*This manuscript presents a wide range of related biochemical and biophysical experiments that present a reasonably self-consistent picture of structure and interactions at the upstream edge of the elongation complex in bacterial RNA polymerase, specifically focusing the relative effects on those measurements of the elongation factor NusG. While any one assay in this toolbox has a range of caveats with respect to the precision of its conclusions, that very different assays present a similar picture provides compelling support for the derived model. The authors are appropriately cautious for the most part, in their broad analyses. For example, fluorescent base analogs are sensitive to more than just base pairing (indeed, they are reflective of pairing-related changes in stacking) and cross linking propensities are are not direct measures of distance.*

*A bigger picture critique might be that these experiments are all carried out on scaffold complexes and so are 'artificial.' Again, the relative consistency of the results assuages such concerns and the power that this approach uniquely yields cannot be ignored. Most time courses are consistent with biologically relevant time scales.*

We agree with the reviewer on the artificial nature of the scaffold system. That said, in this work we use the scaffold system to recreate the effect of NusG previously observed in a conventional promoter initiated TECs. The scaffold setup allows for a more homogeneous preparation and for the modifications that are difficult to introduce into the promoter initiated TECs. Those modifications (unpaired upstream DNA, DNA fluorophores and thiobases) in turn were critical for unraveling the determinants of the anti-backtracking activity of NusG and for the mapping of the upstream edge of the transcription bubble.

*In general, the data analysis in this manuscript is thoughtful and careful. However, in Figure 6, the concentration dependence of kinetic profiles is used to extract forward and reverse rate constants, which are then used to derive binding constants. While the data fit reasonably well, there are some systematic deviations at the extremes. Perhaps more importantly, interdependence of the two fit parameters should be discussed, as they are then used together to calculate Kd and its range. The authors should consider a somewhat more detailed analysis of this fit in the supplementary material. At the very least, they should indicate the equations fit.*

The experiment in Figure 6 Right serves two major purposes:

1) It provides additional evidence that the increase in the TEC17 fluorescence upon the addition of NusG (a small effect often accounting for as little as 20% of the total fluorescence in the sample) is the result of NusG binding: the rates are not too fast and not too slow and the kinetics fits reasonably well to the simple binding scheme.

2) It provides an estimate for the TEC affinity for NusG that in turn reaffirms that our experiments were performed at the saturating concentration of NusG.

In addition, the experiment provides estimates for the rates of NusG binding and dissociation that are tangential to the present investigation but may be ultimately useful in the mathematical modeling of transcription.

The kinetics fits the simple binding scheme reasonably well though there are some deviations at extremes. The deviations may originate from the unaccounted changes in the background fluorescence (in this experiment we operate with a small effect and a large background on the milliseconds timescale) or reflect the additional steps in the binding reaction. For example, the change in the fluorescence following the NusG binding may partially limit the overall reaction rate at high concentration of NusG. However, our analysis indicates that adding extra steps to the simple binding scheme leads to the unconstrained parameters.

We now provided the equations used for the data analysis in Figure 6—figure supplement 2. The equations cannot be explicitly specified in Kintek Explorer but it is implied that they are the simple systems of differential rate equations that are numerically integrated to fit the data.

We now also describe in Figure 6—figure supplement 2 how we accounted for the interdependence of the binding and the dissociation rate constants when determining the upper and lower bounds of the equilibrium dissociation constant of TEC-NusG complex.

*Reviewer #3:*

*[…] 1) I don't buy the argument that the authors have shown that NusG has no effect on steps other than backtracking, in particular on the rate of pause-free transcription elongation. This claim by the authors will need to be revised for the following reasons. First, a previous study that was much more sensitive to small effects that could aggregate across many successive rounds of nucleotide addition reached the opposite conclusion (Herbert et al., 2010), yet the authors do not explain how the present results invalidate these previous results and conclusion. Indeed, their own result shows a modest increase in the overall rate of a single nucleotide turnover reaction even though it is within their experimental error (Figure 2; see bar graphs). Put another way, the modest effects of NusG on pause-free elongation reported previously are not detectable within the experimental error of the present measurements. Further, the authors have looked at only a single template position, whereas the contribution of translocation to elongation and the effect of sequences on upstream fork-junction reannealing and its contribution to translocation and to elongation are certain to vary at different template positions. At least for the basic contribution of translocation to elongation rate, these effects are demonstrated to vary among template positions (e.g., Dangkulwanich et al., 2013 eLife 2: e00971). Given that the authors have provided no evidence to rule out the small effects of NusG previously shown, the authors should modify their conclusion to state that NusG has at most a modest effect on pause-free transcription rate and a greater effect on backtracking, a conclusion indeed essentially the same as that reached by Herbert et al.*

We addressed the reviewer’s request in the following way:

1) We added “measurably” to the sentences about the on-pathway elongation in the abstract and the significance statement. In our view “measurably” is the concise way to state that no effect on the on-pathway elongation was observed in our experiments taking into account the experimental uncertainty.

2) We have rewritten the Discussion as follow:

“Here, we show that NusG slows backtracking but does not affect the on-pathway elongation in the non-paused TEC used in our study. In contrast, (Herbert *et al.,* 2010) concluded that NusG has a modest stimulatory effect (10-20%) on the pause-free elongation rate in addition to inhibiting backtracking. […] In any case, the effect of NusG on the pause-free elongate rate estimated by Herbert et al. is small comparing with the specific effect of NusG on the backtracking rate (~2.5 fold) that we report here.”

We provide in depth discussion on this issue by addressing the summarized queries put forward by the editor (please see above). Most importantly, we argue that the assumption that 1-nt backtracked TECs are uniformly slow and therefore excluded from the pause-free elongation rate range in Herbert et al. is only superficially plausible. It has not been experimentally proven that short backtracking can be equated with pausing. Accordingly, the action of NusG on the pause-free elongation rate may originate from its effect on the 1-nt backtracked TECs that recover with the rates within the pause-free elongation rate range.

Next, we understand the reviewer’s suggestion that the bar graphs in Figure 2 of our original submission possibly showed that NusG has an effect on the half-life of the nucleotide addition cycle that is smaller than our error margins. However, the normalized translocation curves obtained in the parallel experiments with the same TEC preparation +-NusG fully overlap in several independent experiments. Admittedly, the combined analysis of the nucleotide addition and translocation data exaggerated the differences between the total half-lives of the nucleotide addition cycle +-NusG. To address this problem we performed analysis using more datasets and altered the data presentation. Specifically, we now acknowledge that the upper and the lower bounds for the half-lives of the translocation and the nucleotide addition steps should not be treated as independent when estimating the total half-life of the nucleotide addition cycle. The updated Figure 2 provides stronger evidence against the NusG effects on the half-life of the nucleotide addition cycle in our system.

*2) The authors can improve their manuscript by slightly expanding the presentation and discussion of their model for NusG interaction with the upstream fork junction. The authors have appropriately presented this model conservatively given the lack of direct evidence for the NusG-DNA interaction they propose. That said, the impact of the manuscript can be significantly improved by more clearly presenting the model and its implications, as described here and in the next point. First, the authors can much better illustrate their proposed structure for the upstream fork junction. Figure 7 is OK as far as it goes, but it does not aid the viewer in understanding the fork-junction structure nor does it identify the 5 aa loop in NusG proposed to aid in junction reannealing. I suggest the authors prepare additional panels that show two things. In one additional panel, they should show the fork junction from a different perspective so that the partially paired nature of the -10 by and stacking of the -11 bp is more easily understood. Why not use actual base structures for this panel, rather than the simplified stick diagrams that don't really illustrate stacking (or the absence of stacking)? In a second additional panel, the authors should show the sequence of the 5 aa loop and parts of the flanking sheet and helix from E. coli aligned with small number of examples from a diverse evolutionary range of organisms. Please include numbering in this panel so that readers can readily relate the sequences shown to the sequences to full-length NusG/Spt5 sequences.*

We made a multi-panel Figure 8 that illustrates the model and the structural conservation of the five amino acid loop of NusG NTD.

*3) A key question about the authors conclusion is why NusG can inhibit backtracking while having so much less effect on interconversion of the pretranslocated and translocated registers (this discrepancy is equally puzzling whether the effect on the interconversion is modest, as the preponderance of evidence would suggest as described in point 1, or nonexistent, as the authors claim). The authors suggest an entirely reasonable hypothesis that hybrid translocation and downstream DNA translocation may be two distinct, if interconnected, steps, rather than a single translocation event, and that NusG will principally affect events, like backtracking, for which hybrid translocation is rate-limiting). I think this is an important idea, and that the authors could expand their description of it somewhat without crossing the line to overinterpretation of their results. From my perspective, it would be appropriate to say that not only can the preferential effect of NusG on backtracking potentially be explained by the tw0-translocation model, but also that the difficulty of reconciling the large effect of NusG on backtracking and the modest effect on translocation and elongation more generally in the context of a single-step translocation model supports the a two-step translocation mechanism. As long as it's stated this way, and not that the results somehow prove the two-step model, then I think the authors could improve the impact of their manuscript by slightly expanding the discussion of this point.*

We are grateful for the suggestion to expand the implications of our results. We modified the corresponding part of the Discussion. Specifically, we explained the concept of the sequential translocation in more detail and suggested that the specific effect of NusG on backtracking supports a two-step translocation mechanism.